# Complex refractive indices and single scattering albedo of global dust aerosols in the shortwave spectrum and relationship to size and iron content

Claudia Di Biagio[1], Paola Formenti[1], Yves Balkanski[2], Lorenzo Caponi[1,3], Mathieu Cazaunau[1], Edouard Pangui[1], Emilie Journet[1], Sophie Nowak[4], Meinrat O. Andreae[5,6], Konrad Kandler[7], Thuraya Saeed[8], Stuart Piketh[9], David Seibert[10], Earle Williams[11], and Jean–Francois Doussin[1]

[1] LISA, UMR CNRS 7583, Université Paris–Est–Créteil, Université de Paris, Institut Pierre Simon La-place (IPSL), Créteil, France

[2] Laboratoire des Sciences du Climat et de l'Environnement, CEA CNRS UVSQ UP Saclay, 91191, Gif sur Yvette, France

[3] PM_TEN srl, Piazza della Vittoria 7/14, 16121, Genoa, Italy

[4] Plateforme RX UFR de chimie, Université Paris Diderot, Paris, France

[5] Max Planck Institute for Chemistry, P.O. Box 3060, 55020, Mainz, Germany

[6] Geology and Geophysics Department, King Saud University, Riyadh, Saudi Arabia

[7] Institut für Angewandte Geowissenschaften, Technische Universität Darmstadt, Schnittspahnstr. 9, 64287 Darmstadt, Germany

[8] Science Department, College of Basic Education, Public Authority for Applied Education and Training, Al–Ardeya, Kuwait

[9] Climatology Research Group, Unit for Environmental Science and Management, North–West Univer-sity, Potchefstroom, South Africa

[10] Walden University, Minneapolis, Minnesota, USA

[11] Parsons Laboratory, Massachusetts Institute of Technology, Cambridge, Massachusetts, USA

Correspondence to: C. Di Biagio (claudia.dibiagio@lisa.u–pec.fr)

## Abstract

The optical properties of airborne mineral dust depend on its mineralogy, size distribution, and shape, and might vary between different source regions. To date, large differences in refractive index values found in the literature have not been fully explained. In this paper we present a new dataset of complex refractive indices ($m=n-ik$) and single scattering albedos (SSA) for 19 mineral dust aerosols over the 370–950 nm range in dry conditions. Dust aerosols were generated from natural parent soils from eight source regions (Northern Africa, Sahel, Middle East, Eastern Asia, North and South America, Southern Africa, and Australia). They were selected to represent the global scale variability of the dust mineral-ogy. Dust was re–suspended into a 4.2 m³ smog chamber where its spectral shortwave scattering ($\beta_{sca}$) and absorption ($\beta_{abs}$) coefficients, number size distribution, and bulk composition were measured. The complex refractive index was estimated by Mie calculations combining optical and size data, while the spectral SSA was directly retrieved from $\beta_{sca}$ and $\beta_{abs}$ measurements. Dust is assumed to be spherical in the whole data treatment, which introduces a potential source of uncertainty. Our results show that the imaginary part of the refractive index ($k$) and the SSA vary widely from sample to sample, with values for $k$ in the range 0.0011 to 0.0088 at 370 nm, 0.0006 to 0.0048 at 520 nm, and 0.0003 to 0.0021 at 950 nm, and values for SSA in the range 0.70 to 0.96 at 370 nm, 0.85 to 0.98 at 520 nm, and 0.95 to 0.99 at 950 nm. In contrast, the real part of the refractive index ($n$) is mostly source (and wavelength) independent, with an average value between 1.48 and 1.55. The sample–to–sample variability in our dataset of $k$ and SSA is mostly related to differences in the dust's iron content. In particular, a wave-

length–dependent linear relationship is found between the magnitude of $k$ and SSA and the mass con-
centrations of both iron oxide and total elemental iron, with iron oxide better correlated than total ele-
mental iron to both $k$ and SSA. The value of $k$ was found to be independent of size. When the iron oxide
content exceeds 3%, the SSA linearly decreases with increasing fraction of coarse particles at short
wavelengths (< 600 nm).
Compared to the literature, our values for the real part of the refractive index and SSA are in line with
past results, while we found lower values of $k$ compared to most of the literature values currently used
in climate models
We recommend that source–dependent values of the SW spectral refractive index and SSA be used in
models and remote sensing retrievals instead of generic values. In particular, the close relationships
found between $k$ or SSA and the iron content in dust enable establishing predictive rules for spectrally–
resolved SW absorption based on particle composition.

**Introduction**
With teragram amounts of annual emissions, a residence time of about 1–2 weeks in the atmosphere,
and a planetary–scale transport, mineral dust aerosols are a global phenomenon (Uno et al., 2009;
Ginoux et al., 2012), and contribute significantly to the global and regional aerosol loading (Ridley et
al., 2016) and direct radiative effect (Miller et al., 2014).
However, large uncertainties still persist on the magnitude and overall sign of the dust direct radiative
effect (Boucher et al., 2013; Highwood and Ryder, 2014; Kok et al., 2017). One of the major sources of
this uncertainty is our insufficient knowledge of the dust's absorption properties in the shortwave (SW)
and longwave (LW) spectral ranges (e.g., Balkanski et al., 2007; Samset et al., 2018), given that mineral
dust contains large particles and a variety of minerals absorbing over both spectral regions (e.g. iron
oxides, clays, quartz and calcium–rich species; Sokolik and Toon, 1999; Lafon et al., 2006; Di Biagio
et al., 2014a, b). Global and regional scale mapping of dust absorption remains limited and more infor-
mation is required (Samset et al., 2018).
Aerosol absorption is represented both by the imaginary part ($k$) of the complex refractive index ($m=n–$
$ik$) of its constituent material, and by the single scattering albedo (SSA, i.e., the ratio of the scattering
to extinction coefficient) of the particle population, as well as by the mass absorption efficiency (MAE,
units of $m^2$ $g^{-1}$), i.e., the aerosol absorption coefficient per unit mass concentration.
In the shortwave spectral range, absorption by dust accounts for up to ~10–20% of its total extinction.
Dust absorption is highest in the UV–VIS, and almost nil towards the near IR (Cattrall et al., 2003;
Redmond et al., 2010), due to the combined contribution of large particles in the size distribution and
the dust's mineralogy, notably the presence of iron oxides (Karickhoff and Bailey, 1973; Lafon et al.,
2006; Derimian et al., 2008; Moosmüller et al., 2012; Formenti et al., 2014a; 2014b; Engelbrecht et al.,
2016; Caponi et al., 2017). The mineralogy of airborne mineral dust varies according to that of the
parent soils (Nickovic et al., 2012; Journet et al., 2014). Consequently, dust aerosols of different origins
should be more or less absorbing in the SW, and have different imaginary spectral refractive index and
SSA. Field and laboratory measurements, including ground–based and space–borne remote sensing,
show that $k$ varies at a regional scale by almost two orders of magnitude (0.0001–0.008 at 550 nm) with
corresponding SSAs between 0.80 and 0.99 at 550 nm (Volz 1972; Patterson et al., 1977; Shettle and
Fenn, 1979; Dubovik et al., 2002; Haywood et al., 2003; Sinyuk et al., 2003; Linke et al., 2006; Osborne
et al., 2008; Müller et al., 2009; Otto et al., 2009; Petzold et al., 2009; Schladitz et al., 2009; McConnell
et al., 2010; Formenti et al., 2011; Wagner et al., 2012; Ryder et al., 2013a; Engelbrecht et al., 2016;
Rocha–Lima et al., 2018). Albeit some variability being instrumental or analytical (differences in the
sampled size fraction or in the method used to retrieve optical parameters), geographic differences
persist when the same measurement approach and retrieval method are applied, e.g., in AERONET
inversions, supporting the dependence of dust $k$ and SSA with its origin (Dubovik et al., 2002; Koven
and Fung, 2006; Su and Toon, 2011). In contrast, the real part ($n$) of the dust refractive index, mostly
related to particle scattering, is less variable, with values between 1.47–1.56 at 550 nm (e.g., Volz,
1972; Patterson et al., 1977; Balkanski et al., 2007; Petzold et al., 2009).
Differences in $k$ or SSA caused by the spatial variability of the iron content may affect the sign of the
dust radiative effect (heating vs cooling) (Liao and Seinfeld, 1998; Claquin et al. 1999; Miller et al.,
2014), and its global and regional implications (Myhre and Stordal, 2001; Colarco et al., 2014; Das et
al., 2015; Jin et al., 2016; Bangalath and Stenchikov, 2016; Strong et al., 2018). The direct radiative
effect of dust has a strong impact on the Western African Monsoon (Yoshioka et al., 2007; Konaré et
al., 2008) and the Indian Summer Monsoon (Vinoj et al., 2014; Das et al., 2015; Jin et al., 2016). How-
ever, there is no consensus whether dust increases or decreases precipitation over these regions
(Solmon et al., 2008; Jin et al., 2016; Strong et al., 2018). As an example, Solmon et al. (2008) indicate
that dust reduces precipitation over most of the Sahelian region, but increases it over the Northern
Sahel–Southern Sahara. This pattern is, however, very sensitive to the dust absorbing properties, and
a decrease of few percent in dust absorption may even cancel out the increase of precipitation over the
Sahel. Similarly, Jin et al. (2016) show that by varying $k$ from zero to 0.008 at 600 nm (i.e., the highest
value currently used in models) the dust effect on the Indian Summer Monsoon may shift from negative
(reduction of precipitation) to positive (increase of precipitation) values.
In spite of this sensitivity, present climate models adopt a globally–constant spectral complex refractive
index (and SSA) for dust, and hence still implicitly assume the same dust mineralogical composition at
the global scale. This is mainly due to the lack of a globally consistent dataset providing information of
the geographical variability of the dust scattering and absorption properties (e.g., Sunset et al., 2018).
Reference values for the refractive index are usually taken from Volz (1972), Patterson et al. (1977),
D'Almeida et al. (1991), Shettle and Fenn (1979), Sokolik et al. (1993), Sinyuk et al. (2003), or OPAC
(Optical Properties of Aerosols and Clouds, Hess et al., 1998; Koepke et al., 2015). A parameterization
of the spectrally–resolved dust refractive index as a function of the mineralogical composition of the
particles is desirable to replace the globally constant values in current climate models, in particular for
those models that started to incorporate the representation of dust mineralogy into their schemes
(Scanza et al., 2015; Perlwitz et al., 2015a, 2015b).
Improving our knowledge of the spectral SW refractive index of mineral dust and its relation to particle
composition (henceforth origin) is also key for the detection of dust aerosols in the atmosphere and the
quantification of its mass loading, and total or absorption spectral optical depth from active and passive
remote sensing (e.g., Ridley et al., 2016). As an example, the retrieval of the dust SSA and optical depth
over bright desert surfaces with the MODIS (Moderate Imaging Resolution Spectroradiometer) Deep
Blue algorithm (Hsu et al., 2004) applies the Critical Surface Reflectance Method (Kaufman, 1987) to
retrieve dust properties from measured Top of Atmosphere (TOA) spectral reflectance. This algorithm
depends critically on a priori information on the spectral refractive index (Kaufman et al., 2001; Yoshida
et al., 2013). Similarly, active remote sensing techniques (lidar, light detection and ranging) require the
knowledge of the extinction–to–backscatter ratio (the lidar ratio), which is also a strong function of the
complex index of refraction or SSA of the aerosol particles (e.g., Gasteiger et al., 2011; Shin et al.,
2018). Gasteiger et al. (2011) have shown in fact that a 5% change in the SSA at 532 nm can modify
by up to 20% the lidar ratio of dust, which means a 20% change in the estimated profile of the dust
extinction coefficient and retrieved optical depth from lidar measurements.
In this paper we address these issues by reporting of a new laboratory investigation of the shortwave
refractive index and SSA of dust from various source regions worldwide, in the framework of the RED–
DUST project (Di Biagio et al., 2017a; hereafter DB17; Caponi et al., 2017; hereafter C17). Dust optical
properties at discrete wavelengths between 370 and 950 nm are derived in conjunction with the particle
elemental and mineralogical composition, including total elemental iron and iron oxides. We investigate
the relationship of $k$ and SSA to the iron content to provide a parameterization of the dust absorption
as a function of its mineralogy, which can be applied to climate models. The dependence of dust ab-
sorption on the particle coarse size fraction is also investigated to evaluate the change of dust absorp-
tion with atmospheric transport time.
**2. Experimental set–up and instrumentation**
As previously described in DB17 and C17, all experiments discussed here and were conducted in the
4.2 $m^3$ stainless–steel CESAM chamber (French acronym for Experimental Multiphasic Atmospheric
Simulation Chamber) (Wang et al., 2011). Mineral dust aerosols were generated by mechanical shaking
of parent soils using about 15 g of soil sample (first sieved to <1000 μm and then dried at 100 °C)
placed in a 1 L Büchner flask and shaken for about 30 min at 100 Hz by means of a sieve shaker
(Retsch AS200). The dust suspension in the flask was injected into the chamber by flushing with $N_2$ at
10 L $min^{-1}$ for about 10–15 min. After injection in the chamber, the largest fraction of the dust aerosol
(>1.5 μm diameter) remained in suspension for approximately 60 to 120 min thanks to a four–blade
stainless steel fan located at the bottom of the chamber, which also ensured homogeneous conditions
within the chamber volume. The submicron dust fraction, instead, remained constant with time during
the experiments (see Sect. 4.1.1). The evolution of the physico–chemical and optical properties of the
suspended dust was measured by different instruments connected to the chamber. The spectral particle
volume dry scattering ($\beta_{sca}$) and absorption ($\beta_{abs}$) coefficients were measured, respectively, by a 3–
wavelength nephelometer (TSI Inc. model 3563, operating at 450, 550, and 700 nm; 2 L $min^{-1}$ flow rate,
2–s time resolution) and a 7–wavelength aethalometer (Magee Sci. model AE31, operating at 370, 470,

520, 590, 660, 880 and 950 nm; 2 L min$^{-1}$ flow rate, 2–min time resolution). The size distribution of aerosols was measured by means of a scanning mobility particle sizer (SMPS, TSI, DMA Model 3080, CPC Model 3772; mobility diameter range 0.019–0.882 µm; 2.0/0.2 L min$^{-1}$ sheath–aerosol flow rates, 135–s time resolution), a WELAS optical particle counter (OPC) (PALAS, model 2000, white light source between 0.35 and 0.70 µm; optical–equivalent diameter range 0.58–40.7 µm; 2 L min$^{-1}$ flow rate, 1–min time resolution) and a SkyGrimm OPC (Grimm Inc., model 1.129, 0.655 µm operating wavelength; optical–equivalent diameter range 0.25–32 µm; 1.2 L min$^{-1}$ flow rate, 6–s time resolution). Aerosol elemental and mineralogical composition, including iron oxides, was derived by analysis of dust samples collected on polycarbonate filters (47–mm diameter Nuclepore, Whatman, nominal pore size 0.4 µm) mounted in a custom–made stainless–steel sample holder (operated at 6 L min$^{-1}$) for most of the duration of each experiment.

All instruments (size, SW optics, filters) sampled air from the chamber. To equalize the airflow extracted by the different instruments, a particle–free $N_2/O_2$ mixture airflow was continuously injected into the chamber. Inlets for all extractive measurements consisted of a stainless steel tube located inside CESAM, and an external connection of silicone tubing (TSI Inc.) from the chamber to the instruments, for a total length varying between 0.4 and 1.2 m. As detailed in DB17 and shown in Fig. S1 in the supplement, the transmission efficiency due to aspiration and transmission in the sampling lines as a function of particle diameter was estimated to calculate the effective dust fraction sensed by each instrument, taking into account the sampling flow rate, tubing diameter, tubing geometry, and particle shape and density. For the nephelometer and the aethalometer, the length of the sampling line from the intake point in the chamber to the instrument entrance was about 1.2 m, which resulted in a 50% cutoff of the transmission efficiency at 3.9 µm particle geometric diameter and 100% cutoff at 10 µm. For the filter sampling system, the length of the sampling line of about 0.5 m resulted in a 50% (100%) cutoff at 6.5 µm (15 µm) particle geometric diameter (or 50% cutoff at 10.6 µm aerodynamic diameter as indicated in C17, therefore compositional analyses refer to the PM$_{10.6}$ size fraction). For the WELAS, the only OPC considered for size distribution in the coarse fraction (see Sect. 2.2), the 50% (100%) cutoff was reached for particles of 5 µm (8 µm) diameter.

All experiments were conducted at ambient temperature and relative humidity <2%. In addition to overnight evacuation, the chamber was manually cleaned between experiments to avoid contaminations from remaining dust. Background concentrations of aerosols in the chamber were less than 2.0 µg m$^{-3}$ (that is $10^2$ to $10^5$ times smaller than the concentration of dust aerosols in suspension in the chamber during experiments)

A flowchart of the procedure used to treat and combine optical, size, and compositional data, and the algorithm for SSA and complex refractive index retrieval is shown in Fig. 1. Full details of data treatment for size distribution measurements and filter compositional data are provided in DB17 and C17, and in the following we only mention the main points of interest for the present paper. Full details on the data treatment of the SW optical data are provided in Sect. 2.1 and 3.

The optical and size datasets were acquired at different temporal resolutions and then averaged over compatible 10–min intervals, whereas the compositional data represent the experiment integral. The

SSA and complex refractive index data were retrieved both at 10–min resolution and as experiment
averages to relate them to both size and compositional data. Table 1 summarizes the uncertainties on
the measured and derived parameters described in the following.

**2.1 SW optical measurements**

**2.1.1 Aerosol scattering coefficient**

The aerosol scattering coefficients ($\beta_{sca}$) at 450, 550, and 700 nm are measured by the nephelometer
at angles between 7° and 170° and need to be corrected for the restricted field–of–view of the instru-
ment (truncation correction) to retrieve $\beta_{sca}$ at 0°–180°. The truncation correction factor ($C_{trunc}$), i.e., the
ratio of the $\beta_{sca}$ at 0°–180° and 7°–170°, was estimated by Mie calculations for homogeneous spherical
particles using the size distribution measured simultaneously behind SW inlets (see Sect. 2.2). In the
calculations, the real part of the complex refractive index of dust was assumed to be wavelength–inde-
pendent and fixed at a value of 1.53, while the imaginary part was set to 0.003 at 450 and 550 nm and
to 0.001 at 700 nm, according to pre–existing information (Sinyuk et al., 2003; Schladitz et al., 2009;
Formenti et al., 2011; Rocha–Lima et al., 2018). For the different dust samples, $C_{trunc}$ ranged between
1.2 and 1.7 and decreased with wavelength and the dust residence time in the chamber, following the
relative importance of the coarse component in the dust population. The uncertainty on $C_{trunc}$, calculated
by repeating the optical calculations by using the size distribution of dust within its error bars as input
to the optical code, is less than ±5% at all wavelengths. In order to assess the consistency of the derived
truncation correction, we made a sensitivity study in which we recalculated $C_{trunc}$ by varying the refrac-
tive index used as input to the Mie calculations in the range of $n$ and $k$ values obtained in this study
(i.e., values at the 10% and 90% percentile as reported in Table 5 for the whole dataset, that is $n$
between 1.49 and 1.54 and $k$ between 0.001 and 0.006 at 450, 550, and 700 nm). The results of this
sensitivity study indicate that, for fixed dust size distribution, the truncation correction $C_{trunc}$ varies less
than 1% for $n$ between 1.49 and 1.54, and <5% for $k$ between 0.001 and 0.006, and so that it is quite
insensitive to the exact assumed $n$ and $k$ values.
Once corrected for truncation, the spectral $\beta_{sca}$ was extrapolated at the aethalometer wavelengths. With
this aim, the Scattering Ångström Exponents, $SAE_{450–550}$ and $SAE_{550–700}$, were calculated as the linear
fit of $\beta_{sca}$ vs $\lambda$ at 450–550 nm and 550–700 nm, respectively. The $SAE_{450–550}$ and $SAE_{550–700}$ coefficients
were used to extrapolate $\beta_{sca}$ at wavelengths respectively lower and higher than 550 nm. Extrapolated
$\beta_{sca}$ values were used to derive an average SAE of dust for the entire investigated spectral range.

**2.1.2 Aerosol absorption coefficient**

The aerosol absorption coefficient ($\beta_{abs}$) at 370, 470, 520, 590, 660, 880, and 950 nm was retrieved
from aethalometer measurements. The aethalometer measures the attenuation (ATT) through an aer-
osol–laden quartz filter, related to the spectral attenuation coefficient ($\beta_{ATT}$) as:
$$\beta_{ATT}(\lambda) = \frac{\Delta ATT(\lambda)}{\Delta t}\frac{A}{V} \qquad (1)$$

where A is the area of the aerosol collection spot (0.5 ± 0.1) cm$^2$ and V the air sample flow rate (0.002
m$^3$ min$^{-1}$). The slope $\frac{\Delta ATT(\lambda)}{\Delta t}$ is the linear fit of the measured attenuation as a function of time calcu-
lated over 10–min intervals. The spectral attenuation coefficient was converted into an absorption co-
efficient $\beta_{abs}$ following the formula by Collaud Coen et al. (2010):
$$\beta_{abs}(\lambda) = \frac{\beta_{ATT}(\lambda) - \alpha(\lambda)\beta_{sca}(\lambda)}{C_{ref}R(\lambda)} \qquad (2)$$

The $\alpha(\lambda)\beta_{sca}(\lambda)$ term accounts for the fraction of the measured attenuation due to side and backward
scattering and not to light absorption. The Collaud–Coen correction scheme has been recently shown
to yield quite accurate values of the absorption coefficients and absorption Ångström exponents from
aethalometer data (Saturno et al., 2017). The value of $\alpha(\lambda)$ was calculated with the formula by Arnott et
al. (2005) and varied between 0.002 and 0.02 (<±1% from formal error propagation on the Arnott for-
mula), while $\beta_{sca}(\lambda)$ is the scattering coefficient from the nephelometer extrapolated to the aethalometer
wavelengths. The $C_{ref}$ term accounts for multiple scattering by the filter fibers, aerosol laden or not. Its
spectral value, obtained by the linear extrapolation of $C_{ref}$ at 450 and 660 nm estimated for mineral dust
by Di Biagio et al. (2017b), varied between 4.30 at 370 nm and 3.32 at 950 nm. We assume for the
extrapolated $C_{ref}$ an uncertainty of ±10% as estimated in Di Biagio et al. (2017b). The correction factor,
$R$, accounts for the decrease in the aethalometer sensitivity with the increase of the aerosol filter load-
ing. The value of $R$ depends on the absorptivity properties of the sampled aerosol and can be calculated
as a function of the particle SSA. In this study, we calculated $R$ by estimating a first–guess SSA$^*$ as the
ratio of the nephelometer–corrected $\beta_{sca}$ and $\beta_{ext}$ obtained as the sum of $\beta_{sca}$ and the $\beta_{abs}$ non–corrected
for filter loading effect. The $R$ was estimated by using the Collaud–Coen et al. (2010) formulation. For
the range of estimated SSA$^*$ (about 0.60 to 0.99), $R$ varied between 0.5 and 1.0 (±1–10%).
The Absorption Ångstrom Exponent (AAE) was calculated as the power–law fit of $\beta_{abs}$ versus $\lambda$.
Due to an instrumental problem, aethalometer data were not always available, with a typical 30–min
interruption usually 10 to 30 minutes after the beginning of experiments.
**2.2 Size distribution**
The aerosol number size distribution was obtained from SMPS, WELAS and SkyGrimm measurements
over different diameter ranges. The measured electrical mobility and optical equivalent diameters from
the SMPS and the OPCs were first converted into geometrical diameters ($D_g$) as described in DB17
and summarized in Table 1. The OPCs conversion assumes a dust complex refractive index that in our
study was set in the range 1.47–1.53 for $n$ and 0.001–0.005 for $k$ for both the SkyGrimm and the WELAS
(following DB17, for more details see Table 1). After conversion, the estimated $D_g$ range was 0.01–0.50
µm for the SMPS, 0.65–73.0 µm for the WELAS, and 0.29–68.2 µm for the SkyGrimm. Due to a cali-
bration issue, data for the SkyGrimm in the range $D_g$ > 1µm were discarded, so that the WELAS is the
only instrument considered in the super–micron range. A very low counting efficiency was observed for
the WELAS below 1 µm and data in this size range were also discarded.
The SMPS, WELAS, and SkyGrimm data were combined, as detailed in DB17, to obtain the full size
distribution of the dust aerosols suspended in the CESAM chamber, $(dN/dlogD_g)_{CESAM}$, and the size
distribution behind SW optical instruments inlets, $(dN/dlogD_g)_{SWoptics}$, after taking into account particle
losses along sampling lines (see Supplementary material and Fig. S1). As previously discussed, due
to the particle losses in the sampling line from the chamber to the nephelometer/aethalometer, the
$(dN/dlogD_g)_{SWoptics}$ size distribution is cut at 10 microns, so no particles above this diameter reach the
SW instruments.
The measured size distributions, $(dN/dlogD_g)_{CESAM}$ and $(dN/dlogD_g)_{SWoptics}$, were used to estimate the
mass concentration of aerosols and their effective diameter ($D_{eff}$) in the CESAM chamber and behind
the SW instrument inlets as:
$$\text{Mass concentration} = \int \frac{\pi}{6} D_g^3 \frac{dN}{dlogD_g} \rho \cdot dlogD_g \qquad (3)$$

$$D_{eff} = \frac{\int D_g^3 \frac{dN}{dlogD_g} dlogD_g}{\int D_g^2 \frac{dN}{dlogD_g} dlogD_g} \qquad (4)$$

The effective dust density $\rho$ in Eq. (3) was set at 2.5 g cm$^{-3}$, a value that is approximately in the middle
of the range of desert dust densities reported in the literature, i.e., 2.1–2.75 g cm$^{-3}$ (Maring et al., 2000;
Iwasaka et al., 2003; Reid et al., 2003). The effective diameter was evaluated separately for the fine
and coarse fractions of dust by integrating Eq. (4) for diameters ≤1 μm ($D_{eff,fine}$) and >1 μm ($D_{eff,coarse}$),
respectively. For $D_{eff,coarse}$ the upper limit of the calculation is 10 μm when calculated from
$(dN/dlogD_g)_{SWoptics}$, i.e. measured behind the SW inlets.
The dust size distribution, $(dN/dlogD)_{SWoptics}$, measured at each 10–min time step for each sample was
fitted with a sum of five lognormal functions to smooth data inhomogeneities linked to the different
instrument's operating principles and artefacts. Fitting was performed using the Levenberg–Marquardt
algorithm. For each mode, the parameters of the lognormal functions, i.e., the total number concentra-
tion $(N_i)$, the geometric median diameter ($D_{g,i}$), and the geometric standard deviation of the distribution
($\sigma_i$), were retrieved. The uncertainties in the retrieved parameters were estimated by repeating the fit
using size data within their uncertainties. The resulting parameters of the fits at the peak of the injection
in the chamber are reported in Table S1, and an example of size fitting is shown in Fig. S2.
The procedure described here to estimate $(dN/dlogD_g)_{CESAM}$ and $(dN/dlogD_g)_{SWoptics}$ implies that as-
sumptions are made on the values of $n$ and $k$ to correct OPCs data, and this may introduce a circularity
in the estimates of the refractive index of dust that use $(dN/dlogD_g)_{SWoptics}$ as input in optical calculations
(see Sect. 3.2). In order to analyze the dependence of the results on this assumption, we made a
sensitivity calculation by varying the values of $n$ and $k$ used for OPCs corrections within the range of
values retrieved in this study (10% and 90% percentiles in Table 5, i.e., 1.49–1.54 for $n$ and 0.001–
0.006 for $k$). We concluded that changing $n$ and $k$ in this range has a very low impact on the retrieved
number size distribution behind the SW inlets $(dN/dlogD_g)_{SWoptics}$ compared to the original assumptions
made in our calculations (<5% changes in the retrieved size number distribution at the different diame-
ters between the original correction and the correction by varying $n$ and $k$). This is due to the fact that
when changing $D_g$ due to changes in the $n$ and $k$ in the OPCs correction, the loss function also modifies
to values corresponding to the new $D_g$. Given that the loss function increases/decreases for increas-
ing/decreasing $D_g$, the combined changes in $D_g$ and the loss function compensate so that the net num-
ber concentration behind the SW inlets varies less than a few percent. These results therefore suggest
that the procedure to estimate the complex refractive index of dust is nearly independent of the assumed
OPC correction.
Other sources of uncertainties are linked to the spherical assumption to perform the optical to geomet-
rical diameter conversion (discussed in Sect. 3.3) as well as those due to Mie resonance oscillations of
the calculated scattering intensities. Concerning Mie resonances, a sensitivity study was performed
varying the size resolution of our calculations (high/low diameter resolution in the calculations to have
a better/worse reproduction of Mie resonance oscillations) and show that Mie resonances impact the
optical to geometrical correction by less than 1%.
**2.3 Dust elemental and mineralogical composition and iron content**
The elemental and mineralogical composition of the dust aerosols in the $PM_{10.6}$ size fraction was esti-
mated by combining different techniques: X–ray diffraction (XRD, Panalytical model Empyrean diffrac-
tometer) to estimate the particles' mineralogical composition in terms of clays, quartz, calcite, dolomite,
gypsum, and feldspars; wavelength dispersive X–ray fluorescence (WD–XRF, Panalytical PW–2404
spectrometer) to determine the dust elemental composition (Na, Mg, Al, Si, P, K, Ca, Ti, Fe); and X–
ray absorption near–edge structure (XANES) to retrieve the content of iron oxides and their speciation
between hematite and goethite. The dust mass collected on Nuclepore filters during the experiments
varied between 0.3 and 6 mg m$^{-3}$ as calculated from elemental concentrations according to Lide (1992).
Full details on the XRD, WD–XRF, and XANES measurements and data analysis are provided in DB17
and C17. In this study, we discuss the dust elemental iron mass concentration, $MC_{Fe\%}$, i.e., the percent
mass of elemental iron with respect to the total dust mass concentration, and the iron oxides mass
concentration, $MC_{Fe-ox\%}$, i.e., the percent mass fraction of iron oxides with respect to the total dust mass
concentration, estimated as the sum of goethite ($MC_{Goet\%}$) and hematite ($MC_{Hem\%}$) species.
**3. Strategy for data analysis**
**3.1 Calculation of the spectral extinction coefficient and SSA from scattering and absorption**
**coefficients**
The spectral scattering and absorption coefficients, $\beta_{sca}(\lambda)$ and $\beta_{abs}(\lambda)$, measured by the nephelometer
and the aethalometer were used to estimate 10–min averages of the spectral extinction coefficient, $\beta_{ext}$
$(\lambda)$, at the 7–$\lambda$ of the aethalometer between 370 and 950 nm as:
$$\beta_{ext}(\lambda) = \beta_{abs}(\lambda) + \beta_{sca}(\lambda) \qquad (5).$$
The Extinction Ångström Exponent (EAE) was calculated as the power–law fit of $\beta_{ext}$ versus $\lambda$.
The spectral single scattering albedo of dust at 10–min resolution ($SSA_{10-min}$) was retrieved as:
$$\qquad \qquad SSA_{10\text{-min}}(\lambda) = \frac{\beta_{sca}(\lambda)}{\beta_{ext}(\lambda)} \qquad (6).$$
The experiment–averaged SSA (λ) was calculated for each soil type based on the following formula
(Moosmüller et al., 2012):
$$\qquad \qquad SSA(\lambda) = \left(1 + \frac{1}{m(\lambda)}\right)^{-1} \qquad (7)$$
where $m (\lambda)$ represents the slope of the linear fit between the 10–min averages of $\beta_{sca}(\lambda)$ and $\beta_{abs}(\lambda)$
measured along the whole duration of each experiment. An example of $\beta_{sca}(\lambda)$ versus $\beta_{abs}(\lambda)$ fitting to
retrieve the spectral SSA is shown in Fig. S3 in the Supplement. The correlation coefficient $R^2$ of the
$\beta_{sca}(\lambda)$ versus $\beta_{abs}(\lambda)$ fit usually ranges between 0.97 and 1 at all wavelengths. As will be discussed
later in the paper, the single scattering albedo of dust depends on the particle coarse size fraction, and
during our experiments $SSA_{10\text{–min}}$ was not derived continuously for the different samples due to the
aethalometer measurement interruptions. The application of Eq. (7) avoids any bias in the calculated
averaged SSA for different soils due to size effects. For two of the analyzed samples (Tunisia and
Namib–2), however, the linear fitting procedure was not applicable due to the fact that, respectively,
only two and one absorption measurements from the aethalometer were available just after the peak of
the injection, with no data afterwards. Average SSA data for Tunisia were thus estimated as the mean
of the two available $SSA_{10\text{–min}}$ data points, while the single $SSA_{10\text{–min}}$ measurement at the peak of the
injection was reported for Namib–2. This difference in time sampling should be kept in mind when com-
paring data for these two samples to the rest of the dataset.
**3.2 Retrieval of the spectral complex refractive index**
An optical calculation was performed to estimate the complex refractive index ($m=n–ik$) of dust aerosols
based on optical and size data. The retrieval algorithm consisted in recalculating the spectral scattering
$\beta_{sca}(\lambda)$ and absorption $\beta_{abs}(\lambda)$ coefficients measured at each 10–min interval by using the fitted
$(dN/dlogD)_{SWoptics}$ size distribution as input and by varying the real and imaginary parts of the complex
refractive index in the calculations until the best agreement between measurements and calculations
was found. At each wavelength the root mean square deviation (RMSD) was calculated as:
$$RMSD(\lambda) = \sqrt{\left[\frac{\beta_{sca,measured}(\lambda) - \beta_{sca,calculated}(\lambda)(n,k)}{\beta_{sca,calculated}(\lambda)(n,k)}\right]^2 + \left[\frac{\beta_{abs,measured}(\lambda) - \beta_{abs,calculated}(\lambda)(n,k)}{\beta_{abs,calculated}(\lambda)(n,k)}\right]^2} \quad (8)$$
The RMSD was minimized at each wavelength to obtain $n$–$k$ pairs that most closely reproduce the
measured scattering and absorption coefficients. Optical calculations were performed at the 7 wave-
lengths of the aethalometer between 370 and 950 nm using Mie theory. In the calculations, the real part
of the refractive index was varied in the range 1.40–1.60 at steps of 0.01, while the imaginary part was
varied in the range 0.0001–0.050 at steps of 0.0001. For each sample, this resulted in 10500 compu-
tations per wavelength and per 10–min time step. The uncertainty on the real and imaginary parts of
the refractive index was estimated with a sensitivity study. For this purpose, the values of $n$ and $k$ were
also obtained by using as input the observed $\beta_{sca}(\lambda)$, $\beta_{abs}(\lambda)$, and $(dN/dlogD)_{SWoptics}$, plus or minus one
standard deviation on their measurement. The deviations of the values of *n* and k retrieved in the sen-
sitivity study with respect to those obtained in the first inversions were assumed to correspond to the
one standard deviation uncertainty of 10–min retrieved values.
Experiment–averaged values of the spectral *n* and *k* were estimated as the average of single *n* and *k*
values retrieved at 10–min steps (indicated as $n_{10-min}$ and $k_{10-min}$). In fact, differently from the SSA, the
refractive index did not seem to depend on the particle coarse size fraction (Sect. 4.5).
A control experiment was performed with submicron ammonium sulphate aerosols (see DB17 and sup-
plementary Fig. S4) with the aim of validating the proposed methodology to estimate the aerosol com-
plex refractive index for a non–absorbing aerosol type. For ammonium sulphate particles with a mono–
modal size distribution centered at 0.06 μm, as measured with the SMPS, the retrieved real part of the
refractive index was 1.56 (±0.01) in the 450–700 nm wavelength range, as expected from literature
(Toon et al., 1976; Flores et al., 2009; Denjean et al., 2014).

### 3.3 Assumptions on the retrieval of SSA and complex refractive index

The approach used to retrieve the SSA and the complex refractive index of dust and the accuracy of
the results depend on the accuracy of the input data and the assumptions in the optical calculations.
We discuss here two points of the applied procedure, in part already mentioned in the previous para-
graphs.
1/ The size distribution from OPCs and also the scattering coefficient from the nephelometer used as
input to the *n* and *k* retrieval procedure and SSA calculation depend more or less directly on the dust
refractive index. These instruments need in fact to be corrected for instrumental artefacts and these
corrections require an a priori knowledge of the *n* and *k*, which in our approach were set to fixed values
(1.47–1.53 for *n* and 0.001–0.005 for *k* for OPCs optical to geometrical diameter conversion, and 1.53
for *n* and 0.001–0.003 for *k* for nephelometer truncation correction). This choice may in principle intro-
duce a certain degree of uncertainty and circularity into the derived *n*, *k*, and SSA for dust. Nonetheless,
we note that the range of refractive index values used to correct OPCs and nephelometer data falls in
the range of variability of the refractive index values obtained in this study (see Sect. 4.3), which sug-
gests that the values used for the corrections are appropriate. Additionally, as previously discussed,
both the size distribution $(dN/dlogD_g)_{SWoptics}$ and the scattering coefficient are not very sensitive to the
assumptions about *n* and *k* used for the calculations (less than 5% changes in both the number size
distribution behind SW inlets and the scattering coefficient from changing *n* and *k* within the range of
estimated values in this study) which further demonstrates the robustness of the proposed approach.
2/ The retrieval procedure for *n* and *k*, as well as the calculations for OPCs optical–to–geometrical
diameter and the nephelometer truncation correction, simplifies the non–spherical heterogeneous dust
aerosols (e.g., Chou et al. 2008; Okada et al., 2011; Nousiainen and Kandler, 2015) into homogeneous
spherical particles that can be represented by Mie theory. In the present study, we decided not to use
a more advanced shape–representing theory for three main reasons. First, the spherical model has
been shown to produce only moderate errors when computing angular–integrated quantities
(Mishchenko et al., 1995; Otto et al., 2009; Sorribas et al., 2015) such as those we calculate in this

study to retrieve the OPC and truncation corrections and for *n* and *k* retrieval. For instance, Sorribas et al. (2015) showed that using a spheroidal model has a limited effect on the truncation correction. These authors estimated that using a spheroidal model permits to improve by 4 to 13% the agreement between modelled and measured spectral scattering coefficient at 450–700 nm but only for supermicron parti­cles. Conversely, for submicron dust the spherical approximation is better suited than the spheroidal model to reproduce the scattering coefficients by the nephelometer. The study by Mogili et al. (2007) also found an excellent agreement between measured shortwave extinction spectra and those calcu­lated from Mie theory simulations for dust minerals, supporting the use of Mie theory for dust optical modelling. On the other side, other studies point to the need of a non–spherical assumption to improve the modelling of dust optical properties (e.g., Otto et al., 2009). Second, we used Mie theory for the sake of comparison with the large majority of previous field and laboratory data published so far, which had used calculations with the spherical approximation Third, the shape distribution and morphology of the dust samples was not measured during experiments. Improper assumptions on the particle shape and morphology may induce even larger errors than using Mie theory, in particular for super–micron aerosols (Kalashnikova and Sokolik, 2004; Nousiainen and Kandler, 2015). It should be pointed out, however, that dust is usually assumed to be spherical in global climate models (e.g., Myhre and Stordal, 2001; Balkanski et al., 2007; Jin et al., 2016), and different studies still show contradictory results on the true impact of dust non–sphericity on radiative fluxes and heating rates from global model simula­tions (Mishchenko et al., 1995; Yi et al., 2011; Räisänen et al., 2012; Colarco et al., 2014). On the other hand, shape effects can be important for the retrieval of aerosol properties from remote sensing tech­niques using spectral, angular, and polarized reflectance measurements (e.g., Feng et al., 2009). In synthesis, accounting for shape effects is still controversial for dust modelling and also a complex issue beyond the scope of this paper. Thus, while we acknowledge the potential uncertainties induced by spherical assumptions in our study, we do not quantify here the overall impact of this assumption on our results.

## 4. Results

Nineteen soil samples from different desert areas in Northern Africa, Sahel, Eastern Africa and the Middle East, Central Asia, Eastern Asia, North America, South America, Southern Africa, and Australia were selected for experiments from a collection of 137 soil samples from source areas worldwide. The main information on the provenance of these soils is provided in Table 2. The nineteen selected soils, the same as analyzed in DB17, represent the major dust source regions depicted in Ginoux et al. (2012). Amongst the database of 137 samples from all the world regions that constitute significant dust emitters, this range in mineralogical composition represents the largest variability in iron oxides contents that can be found worldwide. This is illustrated in Fig. 2 where we represent the variability of hematite and goe­thite content in the nineteen selected soils and compare it with the range of variability of the global desert soils from the database of Journet et al. (2014).

### 4.1 Physical and chemical properties of analysed dust samples

#### 4.1.1 Dust mass concentration and size distribution

Figure 3 shows a typical example of a time series of aerosol mass concentration and effective fine and
coarse diameters measured inside the CESAM chamber and behind the SW instruments inlets during
the experiments, as well as the corresponding $\beta_{sca}$ and $\beta_{abs}$ at 370 nm. The Figure shows the rapid
increase of the mass concentration within CESAM during dust injection in the chamber, and its subse-
quent decrease during the experiments due to both size–selective gravitational settling, occurring
mostly within the first 30 min of experiments, and dilution by sampling. The scattering and absorption
coefficients of dust decrease with time after injection in tandem with the decrease of the mass concen-
tration and the size–dependent depletion in the chamber. The dust mass concentration inside CESAM
at the peak of the injection is between 2 mg m$^{-3}$ (Mali) and 310 mg m$^{-3}$ (Bodélé) and falls to values
between 0.9 mg m$^{-3}$ (Mali) and 20 mg m$^{-3}$ (Bodélé) behind the SW instruments inlets. These values
are comparable to those measured close to sources during dust storms (Rajot et al., 2008; Kander et
al., 2009). After 2 hours, the dust mass concentration has decreased to values of 0.2 to 2.5 mg m$^{-3}$
(inside CESAM) and of 0.1 to 1.9 mg m$^{-3}$ (behind the SW inlets), as after medium– to long–range dust
transport in the real atmosphere (Weinzerl et al., 2011; Denjean et al., 2016b). This indicates that in a
2–hour experiment in CESAM it is possible to reproduce the temporal changes of the dust mass load
observed in the real atmosphere from emission to medium/long–range transport.
As the mass concentration, the effective diameter of the coarse fraction, $D_{eff,coarse}$, also rapidly de-
creases with time due the progressive deposition of the coarsest particles in the chamber. For the var-
ious soils, $D_{eff,coarse}$ varies in the range of 4–8 µm (peak of injection) to 3–4 µm (after 2 hours) inside the
CESAM chamber, and in the range of 3–4 µm (peak of injection) to 2–3 µm (after 2 hours) behind the
SW inlets. In contrast, $D_{eff,fine}$ remains quite constant during the experiments, with a value between 0.6
and 0.7 µm for all soils. The values of $D_{eff,coarse}$ obtained in this study inside the CESAM chamber are in
line with those measured close to African sources (4–12 µm, Rajot et al., 2008; Weinzerl et al., 2009;
Ryder et al., 2013a) and for dust transported across the Mediterranean (5–8 µm, Denjean et al., 2016a).
Conversely, the values of $D_{eff,coarse}$ behind the SW instruments inlets are mostly in agreement with those
reported for dust transported at Cape Verde and across the Atlantic ocean (~3 µm, Maring et al., 2003;
Müller et al., 2011; Denjean et al., 2016b). Our values of $D_{eff,fine}$ are higher compared to values reported
by Denjean et al. (2016a) for dust aerosols transported over the Mediterranean (0.2 to 0.5 µm), reflect-
ing the fact that we analyse pure dust whereas these authors often encountered dust externally mixed
with pollution particles.
The comparison of $D_{eff,coarse}$ values suggests that while the size distribution in CESAM is mostly repre-
sentative of dust close to sources (see DB17), the size measured behind the SW instruments inlets is
mostly representative of transport conditions. Figure 4 illustrates this point by showing the volume size
distributions of the generated dust aerosols at the peak of injection seen by the SW optical instruments,
compared to the average size of dust measured in CESAM (DB17) and field observations close to
sources (e.g., Niger) and after long–range transport (Cape Verde, Suriname, Puerto Rico, and Barba-
dos). The size distribution of dust inside CESAM includes a coarse mode up to ~ 50 µm and well repro-
duces field observations close to sources, as shown in comparison to the Niger case. Due to particle
losses along tubes, particles above 10 µm diameter are not seen by the SW instruments. The overall
shape of the dust size distribution sensed by the SW instruments is comparable to that measured after
atmospheric long–range transport, even if the fraction of particles above 3.9 µm diameter, which is at
the 50% cutoff of the transmission efficiency for the SW optical instruments, is significantly under–
represented compared to observations (i.e., Betzer et al., 1988; Formenti et al., 2001; Maring et al.,
2003; Ryder et al., 2013b, 2018; Jeong et al., 2014; Denjean et al., 2016b). It should be keep in mind
that often also field data are affected by inlet restrictions so that they cannot measure the whole coarse
dust fraction (see Table 1 in Ryder et al., 2018). The lowest cutoff for field data shown in Fig. 4 are for
the NAMMA and PRIDE datasets and correspond to upper size limits at 5 and 10 µm in diameter,
respectively. Being these values above our cutoff of 3.9 µm, it means that the comparison with our size
dataset is meaningful within the range of our measurements. To note that only the data from AER–D
did not suffer from significant inlet restrictions thus leading to the observation of giant dust particles up
to tens of microns in the Saharan Air Layer off the coasts of Western Africa.
**4.1.2 Iron and iron oxide dust content**
Elemental iron includes the iron in the form of iron oxides and hydroxides, i.e. hematite and goethite
(the so–called free iron, mostly controlling SW absorption) and the iron incorporated in the crystal struc-
ture of silicates and alluminosilicates (illite, smectite), which does not substantially contribute to SW
absorption (Karickhoff and Bailey, 1973; Lafon et al., 2004). The mass concentrations of these compo-
nents (total iron oxides, hematite, goethite, and total elemental iron) for the different analysed samples
are reported in Table 3. There is a considerable variability in the iron and iron oxide content for our
samples. Total iron in the dust samples is in the range from 2.4% (Namib–1) to 10.6% (Namib–2). Iron
oxides account for 11% to 62% of the iron mass (calculated following C17, not reported in Table 3),
whereas the percent of iron oxides to the total dust mass varies between 0.7% (Bodélé Depression)
and 5.8% (Niger). These data are in the range of values reported in the literature (Reid et al., 2003;
Scheuvens et al., 2013; Formenti et al., 2011, 2014a). For the samples from the Sahara and the Sahel,
goethite is the dominant iron oxide species, in agreement with Lafon et al. (2006) and Formenti et al.
(2014a; 2014b). Elsewhere, hematite dominates over goethite, as reported by some studies (Arimoto
et al., 2002; Shen et al., 2006; Lu et al., 2011).
**4.2 Spectral– and time–dependent dust extinction and absorption coefficients, complex refrac-**
**tive index, and SSA**
Figure 5 illustrates a typical spectral– and time–dependent set of measured optical properties. The
spectral extinction coefficient, absorption coefficient, SSA, and real and imaginary parts of the complex
refractive index obtained at 10–min resolution for the Morocco and Algeria samples are shown at the
peak of the dust injection in CESAM and 30 and 90 min after the peak. Figure 5 shows that absorption
decreases with wavelength, but not extinction. The SSA increases from 370 to 590 nm while it is almost
constant between 590 and 950 nm. The imaginary part of the refractive index decreases with λ following
the decrease of $\beta_{abs}$. The real part of the refractive index does not depend on wavelength.
The extinction and absorption coefficients decrease in absolute value with time, as already shown in
Fig. 3. Their spectral dependence remains quite constant with time, but varies from soil to soil. The
experiment–averaged absorption, scattering, and extinction Ångström exponents in the 370–950 nm
spectral range, representing the spectral variation of the absorption, scattering and extinction coeffi-
cients, vary between the values of 1.5 and 2.4 (AAE), −0.4 and 0.4 (SAE), and −0.2 and +0.5 (EAE)
for the different samples. These values are in line with those previously reported by Moosmüller et al.
(2012) and C17 for dust from various locations. The retrieved $n$ and $k$ also show negligible changes of
their spectral shape with time and their magnitude remains approximately constant. In contrast, the SSA
increases with time, in particular below 600 nm wavelength, and its spectral shape changes. This is
mostly due to the decrease of the coarse size fraction with residence time in the chamber, as will be
analysed in Sect. 4.5. Similarly to the absorption, scattering, and extinction coefficients, the spectral
shape of $k$ and SSA is somewhat different between the various samples, with the sharpest spectral
variations observed for the most absorbing samples and a less pronounced spectral variation for the
less absorbing ones, as evident, for example, by comparing the SSA data for Morocco and Algeria in
Fig. 5.
**4.3 Spectral complex refractive index and SSA for the different source regions and comparison**
**to literature data**
Figures 6 and 7 show the experiment–averaged $n$, $k$, and SSA between 370 and 950 nm for the nine-
teen aerosol samples analyzed in this study. Data of $n$, $k$, and SSA and their uncertainties are reported
in Tables 4 and 5 for each sample together with the average values for each of the eight different source
regions and for the full dataset. Figures 6 and 7 show that there are significant differences, both in
magnitude and spectral shape, between the imaginary refractive index and SSA for the different sam-
ples. The highest values of $k$ (0.0048–0.0088 at 370 nm and 0.0012–0.0021 at 950 nm) and lowest
values of SSA (0.70–0.75 at 370 nm and 0.95–0.97 at 950 nm) are obtained for the Niger, Mali, Namib–
2 and Australia samples, which also show the highest values of both the iron oxide content between
3.6% and 5.8% and hematite content between 2.0% and 4.8%. The lowest values ($k$ is 0.0011–0.0012
at 370 and 0.0003–0.0004 at 950 nm, and SSA is in the range 0.91–0.96 at 370 nm and 0.97–0.99 at
950 nm) are obtained for the Bodélé, Namib–1, and Arizona samples, which have iron oxide contents
between 0.7% and 1.5%. Both $k$ and SSA vary from region to region, with the largest absorptions (high-
est $k$, lowest SSA) for the Sahel and Australia and the lowest absorption (lowest $k$, highest SSA) in
North and South America and the Middle East; $k$ and SSA values also vary within the same region, as
illustrated for the Sahelian and Southern African samples. The real part of the refractive index, on the
other hand, is not only almost wavelength–independent, as anticipated, but also relatively invariant from
sample to sample. Its average over the 370–950 nm spectral range is between 1.48 (Gobi) and 1.55
(Ethiopia and Namib–2).
The full envelope of $n$, $k$, and SSA obtained for the entire set of analysed samples is shown in Fig. 8.
The real refractive index is relatively invariant, while the spectral $k$ varies by up to an order of magnitude
(0.001–0.009 at 370 nm and 0.0003–0.002 at 950 nm). The SSA changes accordingly for the different
dust samples at the different wavelengths (30% change at 370 nm corresponding to values between
0.70–0.96 and 4% change at 950 nm for values within 0.95–0.99). The population mean is 1.52 for $n$
(as spectral average) and varies in the range 0.0033–0.0009 for $k$ and 0.85–0.98 for the SSA between
370 and 950 nm (0.0016 and 0.94 as spectral averages for $k$ and SSA) (Fig. 8 and Tables 4 and 5).
The comparison between the full envelope of $n$, $k$, and SSA in this study with literature data is also
shown in Fig. 8. Literature values considered for comparison include estimates from ground–based,
aircraft, and satellite observations, laboratory studies, AERONET inversions, and estimates from mixing
rules based on the dust mineralogical composition. Given that the sample selection in our experiments
fully envelopes the global variability of mineralogy of natural dust, we could expect that our dataset
would also fully envelope the global–scale variability of the dust absorption and scattering properties in
the SW. When comparing with available literature data we found that our $n$ and SSA datasets very well
encompass the range of values indicated in the literature, with only a few outlier points. In contrast, for
the imaginary refractive index the reported range of variability from the literature is significantly larger
than that found in our study, with our range of $k$ being mostly at the lower bound of previous results.
Nonetheless, our range of $k$ values fully envelopes the ensemble of remote sensing and field campaign
data on airborne dust from the previous literature reported in Fig. 8a. The global average spectral values
for $k$ in our study (thick black line) perfectly match the Dubovik et al. (2002) dataset from a synthesis of
AERONET observations from various locations worldwide. Likewise, our $k$ average is also very close
to the dataset by Balkanski et al. (2007), estimated from mineralogical composition assuming 1.5% (by
volume) of hematite in dust, a value shown to allow a reconciliation of climate modelling and satellite
observations of the dust direct SW radiative effect. By comparison, the average dust hematite content
for the ensemble of our analysed samples is 1.8% (in mass), close to the 1.5% value proposed by
Balkanski et al. (2007).
Looking at Fig. 8, the datasets that show the largest values, which also fall outside our estimated range
of $k$ over the entire considered wavelength range are the ones by: (i) Volz (1972), Patterson et al. (1977)
and Hess et al. (1998; i.e., the OPAC 3.1 version database, which is the same $k$ dataset used in the
new OPAC 4.0 version, Koepke et al., 2015) showing larger values than our dataset over the entire
considered wavelength range. These datasets are amongst the most commonly used references for
the dust imaginary refractive index in many climate models; and (ii) the dataset by Wagner et al. (2012)
obtained from laboratory chamber experiments, deviating especially below 600 nm wavelength from
our range of $k$. The reasons for these discrepancies in the $k$ values are difficult to assess, since they
could be related to both instrumental and analytical aspects. In the studies by Volz (1972) and Patterson
et al. (1977), for instance, the complex refractive index was obtained by transmittance and diffuse re-
flectance on pellet samples, a technique that requires the dust to be pressed in a matrix of non–absorb-
ing material. In this case a discrepancy arises from the different optical behaviour between dust com-
pressed in a pellet and the airborne particles. Moreover, Volz (1972) and Patterson et al. (1977) analyse
dust aerosols collected after mid– to long–range transport, thus after the dust has possibly been mixed
with absorbing species.
For the case of Wagner et al. (2012) the imaginary refractive index was retrieved from laboratory cham-
ber experiments on suspended dust, as in our study. Nonetheless, their approach differs in various
aspects from the one applied here and this can lead to the observed differences in the retrieved $k$. First,
the aerosol generation technique is different between the two works and this possibly leads to particles
with different physico–chemical features compared to our study. In Wagner et al. (2012) the dust aerosol
was generated by a rotating brush disperser using only the 20–75 µm sieved fraction of the soils. This
system acts to disaggregate the finest particles of the soil by passing it through a nozzle. Then the
largest aerosol grains were removed by a cyclone system (50% cutoff at 1.2 μm aerodynamic diameter),
so that only the submicron size fraction was measured. We show in Sect. 4.5 that $k$ is independent of
size for the range of investigated effective coarse diameters between 2 and 4 μm, but the range of sizes
analysed in Wagner et al. (2012) is significantly lower than in our study and a size–effect cannot be
excluded. In fact, the relationship between dust absorption and iron content may vary depending on the
considered size fraction (see C17) due to the fact that iron bearing minerals are more concentrated in
the clay fraction (<2.0 μm) of the dust (Kandler et al., 2009). Moreover, generating dust in a different
way may lead to differences in the chemical and mineralogical size–dependent composition of the sam-
ple, therefore contributing to the observed differences. The impact of this is however difficult to evaluate.
Another difference concerns the choice of the optical theory to retrieve k (T–matrix in Wagner et al.
instead of Mie theory as used in our work). This can contribute to the observed differences, even if in a
limited way (Mogili et al., 2007; Sorribas et al., 2015). Third, in their retrieval Wagner et al. fixed the real
refractive index to a wavelength–independent value of 1.53 (as done in several other field and labora-
tory studies in Fig. 8) and this assumption can bias high/low the retrieved $k$ if the actual $n$ is higher/lower
than the assumed 1.53 value. So, in summary, while multiple factors could contribute to the discrepancy
it remains however difficult to assess which source of discrepancy is dominant.

## 4.4 Imaginary refractive index and SSA versus iron and iron oxide content

The sample–to–sample variability of the imaginary part of the refractive index $k$ and the SSA observed
in Fig. 6 and 7 is related to the dust composition by investigating the dependence on the particle iron
content. In Fig. 9 we show the experiment–averaged $k$ and SSA at 370, 520, and 950 nm versus the
mass concentration of iron oxides (hematite+goethite, $MC_{Fe-ox\%}$), hematite ($MC_{Hem\%}$), goethite
($MC_{Goeth\%}$), and total elemental iron ($MC_{Fe\%}$) measured for the different dust samples in this study. The
data are linearly fitted to relate $k$ and SSA to $MC_{Fe-ox\%}$, $MC_{Hem\%}$, $MC_{Goeth\%}$, and $MC_{Fe\%}$. The results of
the fits at all wavelengths between 370 and 950 nm are reported in Table 6, together with the statistical
indicators of the goodness of fit (correlation coefficient, $R^2$, and reduced chi square, $\chi^2_{red}$, i.e., the ob-
tained chi square divided by the number of degrees of freedom). There is an excellent correlation be-
tween both $k$ and SSA and $MC_{Fe-ox\%}$ at the different wavelengths ($R^2$>0.75). A weaker correlation is
found when relating $k$ and SSA to $MC_{Hem\%}$ and $MC_{Fe\%}$ ($R^2$ between 0.40 and 0.74 for $k$ and between
0.49 and 0.78 for the SSA), and $MC_{Goeth\%}$ ($R^2$ between 0.17 and 0.62). The better correlation of $k$ and
SSA to $MC_{Fe-ox\%}$ compared to $MC_{Fe\%}$ is expected since dust optical properties in the visible wavelengths
are mostly sensitive to the fraction of iron oxides, rather than to iron incorporated into the crystal struc-
ture of silicates (Karickhoff and Bailey, 1973; Lafon et al., 2006; Moosmüller et al., 2012; Klaver et al.,
2011; Engelbrecht et al., 2016; C17). The quantities that most robustly satisfy a linear relationship are
$k$ and $MC_{Fe-ox\%}$, as indicated by the reduced chi square $\chi^2_{red}$ that is around 1 at all different wavelengths.
The $\chi^2_{red}$ increases to values also larger than 2 in the other cases, indicating the poorer robustness of
the fit in these cases.
We also investigated the dependence of the spectral $k$ and SSA on the mass concentration of other
minerals, such as clays, calcite, quartz, and feldspars, and also on the mass concentration of different
elements. We found that there is no statistically significant correlation between $k$ or SSA and the mass
concentration of any of these compounds (not shown), with $R^2$ values between 0.002 and 0.46 at the
different wavelengths for all cases.
These results therefore clearly show that iron, particularly in the form of iron oxides (hematite + goe-
thite), is the main driver of dust shortwave absorption. Measuring only the hematite mass fraction to
estimate the dust absorption, as it is sometimes done, is therefore not sufficient.
**4.5 Imaginary refractive index and SSA versus dust coarse size fraction**
The dependence of the spectral $k$ and SSA on the dust coarse fraction is investigated by relating it to
the $D_{eff,coarse}$ calculated from the size distribution data behind the SW instruments inlets. The $k_{10-min}$ and
$SSA_{10-min}$ at 370, 520, and 950 nm versus $D_{eff,coarse}$ are shown in Fig. 10 for all experimental data, which
we separated into three classes based on their iron oxide content ($MC_{Fe-ox\%} \leq 1.5\%$, $1.5\% < MC_{Fe-ox\%}$
$< 3\%$, $MC_{Fe-ox\%} \geq 3\%$). Figure 10 shows that even if the correlation is not very strong ($R^2 < 0.54$), there
is a clearly decreasing tendency for the $SSA_{10-min}$ with increasing $D_{eff,coarse}$, particularly at 370 and 520
nm for strongly absorbing samples with iron oxide content larger than 3%. The $SSA_{10-min}$ is mostly
independent of changes of $D_{eff,coarse}$ at 950 nm. Conversely, $k_{10-min}$ has a very poor correlation with
$D_{eff,coarse}$ ($R^2 < 0.35$) and thus does not depend on size. Similar results were also obtained for the real
part (not shown).
These results confirm previous observations (Sokolik and Toon, 1999; McConnell et al., 2008, 2010;
Ryder et al., 2013a; 2013b) that the refractive index is independent of size. This suggests that size–
dependent mineralogical composition is not sufficient to affect $k$ (in the limit of our measurement and
retrieval procedure precision). It is worth mentioning that only few past studies evidenced a dependence
of $k$ on the size distribution of the dust aerosols (i.e., Kandler et al., 2009, 2011; Otto et al., 2009) maybe
because the refractive index was retrieved in these studies from mixing rules based on the estimated
size–dependent mineralogical composition.
Differently from $k$, the SSA increases as the coarse dust size fraction decreases. This is due to the fact
that absorption efficiency for a single particle ($Q_{abs}$) increases with particle diameter while the scattering
efficiency ($Q_{sca}$) decreases. Ryder et al. (2013a) also showed that the dependence of SSA on size is
linear, but important only when the coarse fraction is high (if particles larger than about 3 µm in diameter
are present), otherwise the SSA depends mainly on composition, also in agreement with more recent
field observations by Ryder et al. (2018).
**5. Summary**
In this paper we presented new measurements of the spectral SW complex refractive index ($m=n–ik$)
and single scattering albedo (SSA) for nineteen mineral dust aerosols generated in the laboratory from
natural soil samples from major desert dust source areas in northern Africa, the Sahel, Middle East,
eastern Asia, North and South America, southern Africa, and Australia, and selected to represent the
heterogeneity of the dust composition at the global scale, in particular the range of iron oxide concen-
trations. The envelope of refractive indices and SSA data obtained in this study can thus be taken as
representative of the variability of the global dust aerosol.
Experiments described here were conducted in the 4.2 m³ CESAM chamber, a dynamic environment
where dust aerosols are generated and maintained in suspension for several hours while monitoring
the evolution of their physical, chemical, and optical properties. The generated dust aerosols are char-
acterized by a realistic size distribution, including both the sub–micron and the super–micron fraction,
and they have an atmospherically representative mass concentration and composition, including iron
oxides and elemental iron content.
Some other laboratory studies have been performed in the past to investigate the shortwave SSA of
dust from different sources worldwide and its dependence on composition (Linke et al., 2006; Moosmül-
ler et al., 2012; Engelbrecht et al., 2016). Conversely, for the refractive index there exists to our
knowledge only one other chamber study (Wagner et al., 2012), that retrieved the imaginary part $k$
between 305 and 955 nm for dust aerosols from a limited number of source areas in Africa (Burkina
Faso, Egypt and Morocco). As a matter of fact, our work provides the first consistent simulation chamber
study of the complex refractive index of global dust.
The results of the present study can be summarized as follows:
1. The spectral $k$ and SSA retrieved in this study vary from sample to sample within the same region
but also from a region to another. For $k$, values vary between 0.0011–0.0088 at 370 nm, 0.0006 to
0.0048 at 520 nm, and 0.0003–0.002 at 950 nm. For SSA, values vary from 0.70 to 0.96 at 370 nm,
0.85 to 0.98 at 520 nm, and from 0.95 at 0.99 at 950 nm. In contrast, $n$ is wavelength–independent
and almost uniform for the different sources, with values between 1.48 and 1.55. Values for $n$ and
SSA fall within the range of published literature estimates, while for $k$ we obtain a much narrower
range of variability than the ensemble of literature results, as illustrated in Fig. 8. In particular, we
found lower values of $k$ compared to most of the literature values currently used in climate models,
such as Volz et al. (1972), Patterson et al. (1977), and the OPAC database (Hess et al., 1998;
Koepke et al., 2015). In their study, Miller et al. (2014) state that the values of Dubovik et al. (2002)
from AERONET, Patterson et al. (1977) for far–travelled dust, and OPAC probably bracket the
global solar absorption by dust. In contrast, our results indicate that dust absorption is lower than
previously thought, and its average is close to the values reported by Dubovik et al. (2002) from
AERONET observations and Balkanski et al. (2007) for a dust with a 1.5% volume fraction of hem-
atite. Our range of variability of an order of magnitude for $k$ and between 4% and 30% for the
spectral SSA is actually large enough to change the sign of the global dust direct effect at the TOA
(Miller et al., 2004), as well as its regional implications (e.g., Solmon et al., 2008; Jin et al., 2016),
and has to be taken into account in climate modelling.
2. The documented changes in $k$ and SSA also impact remote sensing retrievals. To give an example,
following Gasteiger et al. (2011), our observed variability of about 10% for the SSA at 532 nm would
translate to about 40% variability in the retrieved extinction profiles and optical depths from lidar
observations for dust from varying sources.
3. The sample–to–sample variability observed in this study is mostly related to the iron oxide and
elemental iron content in dust. At each investigated wavelength the magnitude of $k$ and SSA is
linearly correlated to the mass concentration of total iron oxides, hematite, goethite, and total ele-
mental iron. Small variations of these compounds translate into large variations of $k$ and SSA.
4. We also investigated the dependence of $k$ and SSA on the size distribution of dust. While $k$ is
independent of size (suggesting that a constant value can be used along transport), below 600 nm
the SSA linearly decreases for increasing $D_{eff,coarse}$ for strongly absorbing samples with more than
3% iron oxide content. The investigated range of $D_{eff,coarse}$ is within about 2 and 4 µm, and thus
comparable to values obtained along a transport path over the Atlantic Ocean for dust during about
2 to 6 days following emission (Denjean et al., 2016a).

5. The observations of points 3 and 4 suggest that while it is sufficient to know the content of iron
oxide (or elemental iron) in dust to predict its spectral $k$, which means that only one tracer is needed
in models to parametrize its regional and global variability, for the spectral SSA both composition
and size distribution are required.

## 6. Concluding remarks

Based on our results, we recommend that dust simulations, as well as remote sensing retrievals, use
source–dependent values of the spectral SW refractive index and SSA instead of generic values. We
propose, as a first step, a set of regionally–averaged $n$, $k$, and SSA values to represent dust from each
of the eight regions analysed here as well as a global average value from the ensemble of our data
(Tables 4 and 5). Furthermore, the relationships found between $k$ and SSA and the iron oxides or ele-
mental iron content in dust open the perspective to establish predictive rules to estimate the spectrally–
resolved SW absorption of dust based on composition. We recommend the use of iron oxide content
rather than iron content as it is better correlated with $k$ and SSA. The relationship found in this study,
nonetheless, refer to the bulk composition of the dust aerosols and to a size range typical of 2 to 6 days
of transport in the atmosphere. As demonstrated in C17 for the mass extinction efficiency, the relation-
ships linking the dust absorption to iron content vary as a function of the analysed size fraction due to
the fact that iron bearing minerals are more concentrated in the clay fraction (<2.0 µm) than in the
coarsest fraction of the dust (Kandler et al., 2009; C17). Further investigation should be therefore ad-
dressed to evaluate the dependence of the spectral $k$ and SSA versus iron content as a function of the
size distribution of the particles, in particular extending to a wider range of $D_{eff,coarse}$ compared to the
one investigated in the present study. This will allow to determine if the $k$ and SSA versus iron relation-
ships change or not in different phases of the aerosol lifetime, so if it is valid close to source areas
(when the coarsest fraction is dominant, i.e. $D_{eff,coarse}$ up to 15 µm, Ryder et al. (2013b)), and in long–
range transport conditions (when most of the coarse particle fraction above few µm has settled out (i.e.,
$D_{eff,coarse}$ of 2–3 µm or lower, Denjean et al. (2016b)).
We point out, however, that the use of mineralogy to estimate $k$ and SSA based on linear relationships,
as obtained in our study, requires that the model–predicted dust composition accurately reflects that of
the natural atmospheric aerosols. To this aim, realistic soil mineralogy databases and accurate model-
ling of the soil to aerosol size fractionation need to be developed in model schemes. In this sense we
mention the EMIT project (Earth Surface Mineral Dust Source Investigation) as a potential near–future
source of high resolution surface mineralogy data for arid and semi–arid regions based on imaging

spectroscopy satellite data (Green et al., 2018). Also, a realistic representation of the size distribution, in particular the coarse mode fraction of dust and its retention during atmospheric transport, should be provided in models given its importance in affecting the SSA, as shown in this study and previously reported in other papers (Ryder et al., 2013a, 2013b, 2018; Kok et al., 2017).

Our study focuses on the dust spectral optical properties between 370 and 950 nm. Further work is required to extend the range of spectral refractive index and SSA data to wavelengths lower than 370 nm or higher than 950 nm given that these data are often required in Global Circulation Models and Numerical Weather Prediction models.

We do not provide any quantification of the uncertainty associated with the assumption of spherical particles in our study, even if we acknowledge the potential role of non–sphericity in affecting our data treatment and results. Additional work is foreseen to better investigate the shape of our generated dust and the impact of non–sphericity on retrieved spectral refractive indices and SSA.

Finally, this study had the objective to investigate the variability of the dust SW optical properties at the global scale linked to the global variability of the dust composition. It is noteworthy that observations over Southern Africa and the Sahel from the present study indicate that the $k$ and SSA variability over these regions is comparable to the one obtained for the global scale. For other regions, such as North America and Australia, only one sample was analyzed, with no information on the regional–scale variability of $k$ and SSA. Additionally, for some of the analyzed areas, such as the Bodélé depression, even local scale variability (on the order of few km) may be of relevance, given the documented local scale changes of the particles' mineralogy and iron content (Bristow et al., 2010). More efforts should be therefore devoted to better characterize the variability of dust spectral optical properties at the regional and sub–regional scale with the aim of better assessing the dust impact on the climate of different areas of the world.

**Data availability**

Complex refractive index and single scattering albedo data for the different analyzed samples are provided in Tables 4 and 5 and will be compiled together with aerosol properties from other studies within the Library of Advanced Data Products (LADP) of the EUROCHAMP datacenter (https://data.eurochamp.org). The CESAM data used in this study are immediately available upon request to the contact author and will also soon be made available through the Database of Atmospheric Simulation Chamber Studies (DASCS) of  the EUROCHAMP datacenter (https://data.eurochamp.org /)).

**Code availability**

The following IDL routines were used in the analysis: mpfitexy.pro (available at https://github.com/williamsmj/mpfitexy) was used to linearly fit data taking into account uncertainties on both x and y; mie_single.pro (available at http://www.atm.ox.ac.uk/code/mie/mie_single.html) was used for optical calculations using Mie theory; mpcurvefit.pro (available at http://cow.physics.wisc.edu/~craigm/idl/idl.html) was used for size lognormal fitting.

**Author contributions**

C. Di Biagio, P. Formenti, Y. Balkanski, and J. F. Doussin designed the experiments and discussed the results. C. Di Biagio performed the experiments and performed the full data analysis with contributions by P. Formenti, L. Caponi, M. Cazaunau, E. Pangui, and J.F. Doussin. The soil samples used for experiments were collected by M. O. Andreae, K. Kandler, T. Saeed, S. Piketh, D. Seibert, and E. Williams. E. Journet participated to the selection of the soil samples for experiments. S. Nowak performed the XRD measurements. C. Di Biagio and P. Formenti wrote the manuscript with comments from all co–authors.

**Acknowledgements**

The RED–DUST project was supported by the French national programme LEFE/INSU and by the OSU–EFLUVE (Observatoire des Sciences de l'Univers–Enveloppes Fluides de la Ville à l'Exobiologie) through dedicated research funding. The authors acknowledge the CNRS–INSU for supporting the CESAM chamber as national facility and the AERIS datacenter ([www.aeris–data.fr](http://www.aeris-data.fr)) for distributing and curing the data produced by the CESAM chamber through the hosting of the EUROCHAMP datacenter. This work has received funding from the European Union's Horizon 2020 research and innovation programme through the EUROCHAMP–2020 Infrastructure Activity under grant agreement no. 730997. C. Di Biagio was supported by the CNRS via the Labex L–IPSL, funded by the ANR (grant no. ANR–10–LABX–0018). K. Kandler is funded by the Deutsche Forschungsgemeinschaft (DFG, German Research Foundation) – 264907654; 416816480 (KA 2280). Field sampling in Saudi Arabia was supported by a grant from King Saud University. The authors thank the LISA staff, who participated in the collection of the soil samples from Tunisia, Niger, Atacama, Patagonia, and the Gobi desert used in this study, and S. Caquineau (LOCEAN), S. Chevaillier (LISA) and G. Landrot (synchrotron SOLEIL), for their contribution in the XRD, WD–XRF and XANES analyses. C. Di Biagio thanks P. Stegmann for providing corrected refractive index data shown in Fig. 8. The authors wish also to acknowledge C. Ryder and C. Pérez Garcia Pando for providing valuable comments that helped to increase the readability and quality of the paper.

**Competing interests**

The authors declare that they have no conflict of interest.

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

**Table 1.** Measured and retrieved quantities and their estimated relative uncertainties. For further details, refer to Sect. 2, as well DB17 and C17.

| Parameter | | Time resolution | Relative uncertainty | Uncertainty calculation | Comments |
|---|---|---|---|---|---|
| Optical SW | Scattering coefficient at 450, 550, and 700 nm, $\beta_{sca}$ ($\lambda$) | *10–min data* | 5–12% | Quadratic combination of photon counting and gas calibration uncertainty (5%), angular corrections uncertainty (<5%) and standard deviation over 10–min intervals (2–10%). | The uncertainty on $\beta_{sca}$ ($\lambda$) usually decreases with increasing dust residence time in the chamber as a result of the reduction of the coarse component. |
| | Absorption coefficient at 370, 470, 520, 590, 660, 880, and 950 nm, $\beta_{abs}$ ($\lambda$) | *10–min data* | 22–30% at 370 nm 23–87% at 950 nm | Error propagation formula[1] on Eq. (2) considering the uncertainties on $\beta_{ATT}(\lambda)$ from 10–min fitting procedure (error propagation formula[1] on Eq. 1, ~20%), and uncertainties on $\alpha(\lambda)$ (1%), $\beta_{sca}(\lambda)$ (5–12%), $C_{ref}$ (10%), and R (1–10%). | |
| | Extinction coefficient, $\beta_{ext}$ ($\lambda$) = $\beta_{sca}$ ($\lambda$) + $\beta_{abs}$ ($\lambda$) | *10–min data* | ~25% | Sum of $\beta_{sca}$ ($\lambda$) and $\beta_{abs}$ ($\lambda$) uncertainties | |
| | Single Scattering Albedo, SSA ($\lambda$) = $\beta_{sca}$ ($\lambda$) / ($\beta_{sca}$ ($\lambda$) + $\beta_{abs}$ ($\lambda$)) | *10–min data* | 9–12% | Error propagation formula[1] considering single uncertainties on $\beta_{sca}(\lambda)$ and $\beta_{abs}(\lambda)$. | |
| | Single Scattering Albedo, SSA ($\lambda$) = $(1+1/m(\lambda))^{-1}$ | *Experiment averaged* | 1–12% at 370 nm 1–3% at 950 nm | Error propagation formula[1] on Eq. (6) considering the uncertainty on m($\lambda$), i.e., the slope of the linear fit between $\beta_{sca}$ ($\lambda$) and $\beta_{abs}$ ($\lambda$) over the whole duration of each experiment. | |
| | Complex refractive index ($n$–i$k$) | *10–min data* | <5% for $n$ <50% for $k$ | Deviations of the values of $n$ and $k$ retrieved in the sensitivity study (see Sect. 3.2) with respect to those obtained in the first inversions were assumed to correspond to the one standard deviation uncertainty to 10–min retrieved values. | |
| | Complex refractive index ($n$–i$k$) | *Experiment averaged* | <8% for $n$ 13–75 % for $k$ | Quadratic combination of the standard deviation of $n$ and $k$ over the experiment and the deviation on the experiment–averaged values between those obtained from central inversions and inversions using input data ± their uncertainty. | |
| Size distribution | SMPS geometrical diameter ($D_g$), $D_g = D_m / \chi$ | – | ~6% | Error propagation formula[1] considering the uncertainty on the estimated shape factor $\chi$ (~6%) | The electrical mobility to geometrical diameter conversion was performed by assuming for dust a dynamic shape factor of 1.75 ± 0.10, as determined by SMPS–SkyGrimm comparison in their overlapping range (see DB17) |

| | | | | | |
|---|---|---|---|---|---|
| | SkyGrimm geometrical diameter ($D_g$) | – | <15.2% | Standard deviation of the $D_g$ values obtained for different refractive indices values used in the optical to geometrical conversion | The conversion of optical to geometrical diameters for the SkyGrimm and the WELAS was performed by taking into account the visible complex refractive index of dust aerosols. Optical calculations were computed at the SkyGrimm operating wavelength (0.655 μm) and over the spectral range of the WELAS (0.35 to 0.7 μm) using Mie theory for spherical particles by fixing *n* at 1.47, 1.50, and 1.53, and by varying *k* in steps of 0.001 between 0.001 and 0.005. Then $D_g$ is set at the mean ± 1 standard deviation of the values obtained for the different values of *n* and *k* (see DB17). Refractive index is assumed to be constant with particle size and wavelength–independent. |
| | WELAS geometrical diameter ($D_g$) | – | <7% | The same as for the SkyGrimm | |
| | *(dN/dlogD)$_{SWoptics}$* | *10–min data* | ~20–90% | Error propagation formula[1] considering the *dN/dlogD$_g$* st. dev. over 10–min and the uncertainty on particle loss function along sampling tubes *L(D$_g$)* (~50% at 2 μm, ~10% at 8 μm) | The uncertainty of *L(D$_g$)* was estimated with a sensitivity study by varying the values of the input parameters to the Particle Loss Calculator software within their uncertainties (see DB17) |
| | $D_{eff,fine}$ | *10–min data* | <5% | Deviation obtained by repeating the calculations by using the size distribution ± its uncertainty. | |
| | $D_{eff,coarse}$ | *10–min data* | 5–40% | | |
| Mineralogical composition | Elemental iron mass concentration ($MC_{Fe\%}$) | *Experiment averaged* | 10% | Uncertainties calculated as discussed in DB17 and C17 | |
| | Iron oxides mass concentration ($MC_{Fe–ox\%}$) | *Experiment averaged* | 15% | | |
| | Goethite mass concentration ($MC_{Goet\%}$) | *Experiment averaged* | <10% | | |
| | Hematite mass concentration ($MC_{Hem\%}$) | *Experiment averaged* | <10% | | |

[1] $\sigma_f = \sqrt{\sum_{i=1}^{n}\left(\dfrac{\partial f}{\partial x_i}\sigma_{x_i}\right)^2}$










**Table 2.** Summary of information on the soil samples and sediments used in this study.

| Geographical area | Sample | Coordinates | Desert area |
|---|---|---|---|
| Northern Africa – Sahara | Tunisia | 33.02°N, 10.67°E | Maouna |
| | Morocco | 31.97°N, 3.28°W | east of Ksar Sahli |
| | Libya | 27.01°N, 14.50°E | Sebha |
| | Algeria | 23.95°N, 5.47°E | Ti–n–Tekraouit |
| | Mauritania | 20.16°N, 12.33°W | east of Aouinet Nchir |
| Sahel | Niger | 13.52°N, 2.63°E | Banizoumbou |
| | Mali | 17.62°N, 4.29°W | Dar el Beida |
| | Bodélé | 17.23°N, 19.03°E | Bodélé depression |
| Eastern Africa and the Middle East | Ethiopia | 7.50°N, 38.65°E | Lake Shala National Park |
| | Saudi Arabia | 27.49°N, 41.98°E | Nefud |
| | Kuwait | 29.42°N, 47.69°E | Kuwaiti |
| Eastern Asia | Gobi | 39.43°N, 105.67°E | Gobi |
| | Taklimakan | 41.83°N, 85.88°E | Taklimakan |
| North America | Arizona | 33.15 °N, 112.08°W | Sonoran |
| South America | Atacama | 23.72°S, 70.40°W | Atacama |
| | Patagonia | 50.26°S, 71.50°W | Patagonia |
| Southern Africa | Namib–1 | 21.24°S, 14.99°E | Namib |
| | Namib–2 | 19.00°S, 13.00°E | Namib |
| Australia | Australia | 31.33°S, 140.33°E | Strzelecki |


**Table 3**. Chemical characterization of the dust aerosols in the $PM_{10.6}$ size fraction. Column 3 shows
$MC_{Fe\%}$, the fractional mass of elemental iron with respect to the total dust mass concentration (±10%
relative uncertainty), and column 4 reports $MC_{Fe–ox\%}$, the mass fraction of iron oxides with respect to the
total dust mass concentration (±15% relative uncertainty) and its speciation in hematite $MC_{Hem\%}$ and
goethite $MC_{Goeth\%}$ (<±10% relative uncertainty). The iron oxide measurements were not made on the
Taklimakan sample. Mean values and standard deviations based on single sample data are reported
for the full dataset.

| Geographical area | Sample | $MC_{Fe\%}$ | $MC_{Fe–ox\%}$ | $MC_{Hem\%}$ | $MC_{Goet\%}$ |
|---|---|---|---|---|---|
| Northern Africa – Sahara | Tunisia | 4.1 | 2.2 | 1.2 | 1.1 |
| | Morocco | 3.6 | 1.4 | 0.4 | 1.0 |
| | Libya | 5.2 | 3.1 | 0.9 | 2.2 |
| | Algeria | 6.6 | 2.7 | 1.4 | 1.4 |
| | Mauritania | 8.1 | 3.3 | 3.3 | 0.0 |
| Sahel | Niger | 6.1 | 5.8 | 2.3 | 3.5 |
| | Mali | 6.6 | 3.7 | 2.0 | 1.7 |
| | Bodélé | 4.1 | 0.7 | 0.7 | 0.0 |
| Eastern Africa and the Middle East | Ethiopia | 6.8 | 2.0 | 2.0 | 0.0 |
| | Saudi Arabia | 3.8 | 2.6 | 1.8 | 0.8 |
| | Kuwait | 5.0 | 1.5 | 1.5 | 0.0 |
| Eastern Asia | Gobi | 4.8 | 0.9 | 0.9 | 0.0 |
| | Taklimakan | 5.8 | – | – | – |
| North America | Arizona | 5.3 | 1.5 | 1.5 | 0.0 |
| South America | Atacama | 4.7 | 1.6 | 1.6 | 0.0 |
| | Patagonia | 5.1 | 1.5 | 0.9 | 0.6 |
| Southern Africa | Namib–1 | 2.4 | 1.1 | 0.8 | 0.3 |
| | Namib–2 | 10.6 | 4.8 | 4.8 | 0.0 |
| Australia | Australia | 7.2 | 3.6 | 3.6 | 0.0 |
| **Full dataset mean (st. dev.)** | | **5.6 (1.9)** | **2.4 (1.4)** | **1.8 (1.1)** | **0.7 (1.0)** |

Table 4. Real (*n*) and imaginary (*k*) parts of the refractive index estimated for the nineteen analysed dust samples and mean values calculated for the eight regions and for the full dataset. Data for single soils are reported as experiment–averaged values and their uncertainty is calculated as indicated in Table 1. Mean values and standard deviations at each wavelength based on single sample data are reported for the eight regions and the full dataset. The median and 10% and 90% percentile values are also reported for the full dataset. For North America and Australia, for which only one dust sample was analysed, the reported data correspond to the single sample available from these regions. For the real part, the average over the whole shortwave range ($n_{SW}$) is indicated.

| Sample/Region | $n_{SW}$ 037–0.95 µm | $\sigma_{nSW}$ 037–0.95 µm | *k* 0.37 µm | *k* 0.47 µm | *k* 0.52 µm | *k* 0.59 µm | *k* 0.66 µm | *k* 0.88 µm | *k* 0.95 µm | $\sigma_k$ 0.37 µm | $\sigma_k$ 0.47 µm | $\sigma_k$ 0.52 µm | $\sigma_k$ 0.59 µm | $\sigma_k$ 0.66 µm | $\sigma_k$ 0.88 µm | $\sigma_k$ 0.95 µm |
|---|---|---|---|---|---|---|---|---|---|---|---|---|---|---|---|---|
| Tunisia | 1.51 | 0.06 | 0.0045 | 0.0035 | 0.0026 | 0.0018 | 0.0015 | 0.0013 | 0.0012 | 0.0030 | 0.0026 | 0.0018 | 0.0012 | 0.0010 | 0.0008 | 0.0007 |
| Morocco | 1.49 | 0.03 | 0.0023 | 0.0016 | 0.0012 | 0.0008 | 0.0007 | 0.0006 | 0.0007 | 0.0006 | 0.0004 | 0.0003 | 0.0002 | 0.0002 | 0.0002 | 0.0002 |
| Lybia | 1.5 | 0.04 | 0.0029 | 0.0019 | 0.0014 | 0.0007 | 0.0006 | 0.0007 | 0.0007 | 0.0006 | 0.0004 | 0.0002 | 0.0001 | 0.0002 | 0.0002 | 0.0002 |
| Algeria | 1.52 | 0.04 | 0.0025 | 0.0016 | 0.0012 | 0.0007 | 0.0005 | 0.0006 | 0.0006 | 0.0010 | 0.0006 | 0.0004 | 0.0003 | 0.0003 | 0.0003 | 0.0003 |
| Mauritania | 1.5 | 0.03 | 0.0043 | 0.0033 | 0.0026 | 0.0014 | 0.0013 | 0.0010 | 0.0010 | 0.0010 | 0.0009 | 0.0008 | 0.0003 | 0.0003 | 0.0004 | 0.0003 |
| **Northern Africa – Sahara (mean and st. dev.)** | **1.51** | **0.03** | **0.0033** | **0.0024** | **0.0018** | **0.0011** | **0.0009** | **0.0008** | **0.0008** | **0.0010** | **0.0010** | **0.0007** | **0.0005** | **0.0004** | **0.0003** | **0.0003** |
| Niger | 1.51 | 0.04 | 0.0088 | 0.0061 | 0.0048 | 0.0034 | 0.0031 | 0.0028 | 0.0021 | 0.0043 | 0.0031 | 0.0023 | 0.0018 | 0.0015 | 0.0010 | 0.0013 |
| Mali | 1.52 | 0.05 | 0.0048 | 0.0038 | 0.0030 | 0.0023 | 0.0024 | 0.0021 | 0.0021 | 0.0008 | 0.0006 | 0.0004 | 0.0003 | 0.0003 | 0.0003 | 0.0003 |
| Bodélé | 1.49 | 0.03 | 0.0011 | 0.0007 | 0.0006 | 0.0004 | 0.0004 | 0.0003 | 0.0003 | 0.0006 | 0.0004 | 0.0003 | 0.0002 | 0.0002 | 0.0001 | 0.0001 |
| **Sahel (mean and st. dev.)** | **1.51** | **0.03** | **0.0049** | **0.0035** | **0.0028** | **0.0020** | **0.0020** | **0.0017** | **0.0015** | **0.0038** | **0.0027** | **0.0021** | **0.0015** | **0.0014** | **0.0013** | **0.0011** |
| Ethiopia | 1.55 | 0.06 | 0.0026 | 0.0020 | 0.0016 | 0.0013 | 0.0011 | 0.0007 | 0.0006 | 0.0009 | 0.0008 | 0.0007 | 0.0005 | 0.0004 | 0.0002 | 0.0002 |
| Saudi Arabia | 1.54 | 0.06 | 0.0028 | 0.0021 | 0.0015 | 0.0007 | 0.0006 | 0.0006 | 0.0006 | 0.0006 | 0.0005 | 0.0004 | 0.0002 | 0.0001 | 0.0001 | 0.0001 |
| Kuwait | 1.50 | 0.04 | 0.0016 | 0.0010 | 0.0008 | 0.0006 | 0.0005 | 0.0005 | 0.0004 | 0.0005 | 0.0003 | 0.0003 | 0.0002 | 0.0002 | 0.0003 | 0.0002 |
| **Eastern Africa and the Middle East (mean and st. dev.)** | **1.53** | **0.05** | **0.0023** | **0.0017** | **0.0013** | **0.0009** | **0.0007** | **0.0006** | **0.0005** | **0.0007** | **0.0006** | **0.0005** | **0.0004** | **0.0003** | **0.0001** | **0.0001** |
| Gobi | 1.48 | 0.05 | 0.0041 | 0.0025 | 0.0018 | 0.0012 | 0.0011 | 0.0012 | 0.0012 | 0.0017 | 0.0009 | 0.0006 | 0.0004 | 0.0004 | 0.0005 | 0.0005 |
| Taklimakan | 1.54 | 0.07 | 0.0018 | 0.0012 | 0.0009 | 0.0006 | 0.0005 | 0.0005 | 0.0005 | 0.0008 | 0.0005 | 0.0004 | 0.0002 | 0.0002 | 0.0002 | 0.0002 |
| **Eastern Asia (mean and st. dev.)** | **1.51** | **0.05** | **0.0030** | **0.0019** | **0.0014** | **0.0009** | **0.0008** | **0.0008** | **0.0009** | **0.0016** | **0.0009** | **0.0006** | **0.0005** | **0.0005** | **0.0005** | **0.0005** |
| Arizona | 1.51 | 0.05 | 0.0011 | 0.0009 | 0.0007 | 0.0005 | 0.0005 | 0.0005 | 0.0004 | 0.0005 | 0.0004 | 0.0003 | 0.0002 | 0.0002 | 0.0002 | 0.0002 |
| **North America (mean and st. dev.)** | **1.51** | **0.05** | **0.0011** | **0.0009** | **0.0007** | **0.0005** | **0.0005** | **0.0005** | **0.0004** | **0.0005** | **0.0004** | **0.0003** | **0.0002** | **0.0002** | **0.0002** | **0.0002** |
| Atacama | 1.54 | 0.07 | 0.0016 | 0.0015 | 0.0012 | 0.0008 | 0.0006 | 0.0006 | 0.0006 | 0.0005 | 0.0004 | 0.0003 | 0.0002 | 0.0002 | 0.0002 | 0.0002 |
| Patagonia | 1.53 | 0.07 | 0.0024 | 0.0016 | 0.0011 | 0.0009 | 0.0006 | 0.0007 | 0.0006 | 0.0008 | 0.0005 | 0.0003 | 0.0003 | 0.0003 | 0.0003 | 0.0002 |
| **South America (mean and st. dev.)** | **1.54** | **0.06** | **0.0020** | **0.0015** | **0.0011** | **0.0008** | **0.0006** | **0.0007** | **0.0006** | **0.0006** | **0.0001** | **0.0001** | **0.0001** | **0.0000** | **0.0001** | **0.0000** |
| Namib–1 | 1.53 | 0.06 | 0.0012 | 0.0009 | 0.0006 | 0.0004 | 0.0003 | 0.0004 | 0.0004 | 0.0006 | 0.0004 | 0.0003 | 0.0002 | 0.0001 | 0.0002 | 0.0001 |
| Namib–2 | 1.55 | 0.07 | 0.0072 | 0.0054 | 0.0044 | 0.0025 | 0.0018 | 0.0014 | 0.0014 | 0.0027 | 0.0019 | 0.0016 | 0.0009 | 0.0007 | 0.0006 | 0.0006 |
| **Southern Africa (mean and st. dev.)** | **1.54** | **0.06** | **0.0042** | **0.0031** | **0.0025** | **0.0014** | **0.0011** | **0.0009** | **0.0009** | **0.0042** | **0.0032** | **0.0027** | **0.0015** | **0.0010** | **0.0007** | **0.0007** |
| Australia | 1.54 | 0.06 | 0.0058 | 0.0042 | 0.0033 | 0.0017 | 0.0013 | 0.0013 | 0.0012 | 0.0022 | 0.0011 | 0.0010 | 0.0006 | 0.0006 | 0.0004 | 0.0003 |
| **Australia (mean and st. dev.)** | **1.54** | **0.06** | **0.0058** | **0.0042** | **0.0033** | **0.0017** | **0.0013** | **0.0013** | **0.0012** | **0.0022** | **0.0011** | **0.0010** | **0.0006** | **0.0006** | **0.0004** | **0.0003** |
| | | | | | | | | | | | | | | | | |
| **Full dataset (mean and st. dev.)** | **1.52** | **0.04** | **0.0033** | **0.0024** | **0.0018** | **0.0012** | **0.0010** | **0.0009** | **0.0009** | **0.0021** | **0.0016** | **0.0013** | **0.0008** | **0.0007** | **0.0006** | **0.0005** |
| **Full dataset median** | **1.52** | | **0.0026** | **0.0019** | **0.0014** | **0.0008** | **0.0006** | **0.0007** | **0.0006** | | | | | | | |
| **Full dataset 10% percentile** | **1.49** | | **0.0012** | **0.0009** | **0.0007** | **0.0005** | **0.0004** | **0.0004** | **0.0004** | | | | | | | |
| **Full dataset 90% percentile** | **1.54** | | **0.0061** | **0.0044** | **0.0035** | **0.0023** | **0.0019** | **0.0015** | **0.0015** | | | | | | | |

**Table 5.** As in Table 4 for the single scattering albedo (SSA) data.

| Sample/Region | SSA | | | | | | | $\sigma_{SSA}$ | | | | | | |
|---|---|---|---|---|---|---|---|---|---|---|---|---|---|---|
| | 0.37 μm | 0.47 μm | 0.52 μm | 0.59 μm | 0.66 μm | 0.88 μm | 0.95 μm | 0.37 μm | 0.47 μm | 0.52 μm | 0.59 μm | 0.66 μm | 0.88 μm | 0.95 μm |
| Tunisia | 0.85 | 0.90 | 0.93 | 0.95 | 0.95 | 0.97 | 0.97 | 0.03 | 0.02 | 0.02 | 0.01 | 0.01 | 0.01 | 0.01 |
| Morocco | 0.92 | 0.95 | 0.96 | 0.98 | 0.98 | 0.98 | 0.99 | 0.01 | 0.01 | 0.01 | 0.00 | 0.00 | 0.00 | 0.00 |
| Lybia | 0.89 | 0.93 | 0.95 | 0.98 | 0.98 | 0.98 | 0.98 | 0.02 | 0.01 | 0.01 | 0.00 | 0.00 | 0.00 | 0.00 |
| Algeria | 0.87 | 0.92 | 0.94 | 0.97 | 0.97 | 0.98 | 0.98 | 0.02 | 0.01 | 0.01 | 0.00 | 0.00 | 0.00 | 0.00 |
| Mauritania | 0.85 | 0.90 | 0.94 | 0.96 | 0.97 | 0.98 | 0.98 | 0.02 | 0.01 | 0.01 | 0.01 | 0.01 | 0.00 | 0.00 |
| **Northern Africa – Sahara (mean and st. dev.)** | **0.88** | **0.92** | **0.94** | **0.97** | **0.97** | **0.98** | **0.98** | **0.03** | **0.02** | **0.02** | **0.01** | **0.01** | **0.01** | **0.01** |
| Niger | 0.72 | 0.85 | 0.89 | 0.91 | 0.92 | 0.94 | 0.95 | 0.09 | 0.09 | 0.07 | 0.05 | 0.05 | 0.03 | 0.02 |
| Mali | 0.75 | 0.85 | 0.89 | 0.93 | 0.95 | 0.96 | 0.96 | 0.04 | 0.03 | 0.02 | 0.02 | 0.02 | 0.01 | 0.01 |
| Bodélé | 0.96 | 0.98 | 0.98 | 0.99 | 0.99 | 0.99 | 0.99 | 0.04 | 0.02 | 0.02 | 0.01 | 0.01 | 0.01 | 0.01 |
| **Sahel (mean and st. dev.)** | **0.81** | **0.89** | **0.92** | **0.94** | **0.95** | **0.96** | **0.97** | **0.13** | **0.07** | **0.05** | **0.04** | **0.04** | **0.03** | **0.02** |
| Ethiopia | 0.80 | 0.86 | 0.90 | 0.92 | 0.94 | 0.97 | 0.97 | 0.03 | 0.03 | 0.02 | 0.02 | 0.01 | 0.01 | 0.01 |
| Saudi Arabia | 0.88 | 0.93 | 0.96 | 0.98 | 0.98 | 0.98 | 0.98 | 0.03 | 0.02 | 0.01 | 0.01 | 0.01 | 0.00 | 0.00 |
| Kuwait | 0.95 | 0.97 | 0.98 | 0.98 | 0.99 | 0.99 | 0.99 | 0.02 | 0.01 | 0.01 | 0.01 | 0.01 | 0.01 | 0.00 |
| **Eastern Africa and the Middle East (mean and st. dev.)** | **0.88** | **0.92** | **0.94** | **0.96** | **0.97** | **0.98** | **0.98** | **0.07** | **0.05** | **0.04** | **0.03** | **0.03** | **0.01** | **0.01** |
| Gobi | 0.88 | 0.92 | 0.94 | 0.96 | 0.97 | 0.97 | 0.97 | 0.04 | 0.03 | 0.02 | 0.01 | 0.01 | 0.01 | 0.01 |
| Taklimakan | 0.82 | 0.88 | 0.92 | 0.95 | 0.96 | 0.96 | 0.96 | 0.03 | 0.02 | 0.02 | 0.01 | 0.01 | 0.01 | 0.01 |
| **Eastern Asia (mean and st. dev.)** | **0.85** | **0.90** | **0.93** | **0.96** | **0.96** | **0.97** | **0.97** | **0.04** | **0.03** | **0.02** | **0.01** | **0.01** | **0.01** | **0.01** |
| Arizona | 0.93 | 0.96 | 0.97 | 0.98 | 0.98 | 0.99 | 0.99 | 0.01 | 0.01 | 0.01 | 0.00 | 0.00 | 0.00 | 0.00 |
| **North America (mean and st. dev.)** | **0.93** | **0.96** | **0.97** | **0.98** | **0.98** | **0.99** | **0.99** | **0.01** | **0.01** | **0.01** | **0.00** | **0.00** | **0.00** | **0.00** |
| Atacama | 0.89 | 0.93 | 0.94 | 0.97 | 0.97 | 0.98 | 0.98 | 0.03 | 0.02 | 0.02 | 0.01 | 0.01 | 0.01 | 0.01 |
| Patagonia | 0.88 | 0.91 | 0.94 | 0.96 | 0.97 | 0.98 | 0.98 | 0.02 | 0.02 | 0.01 | 0.01 | 0.01 | 0.00 | 0.01 |
| **South America (mean and st. dev.)** | **0.89** | **0.92** | **0.94** | **0.96** | **0.97** | **0.98** | **0.98** | **0.00** | **0.01** | **0.00** | **0.00** | **0.00** | **0.00** | **0.00** |
| Namib–1 | 0.91 | 0.95 | 0.96 | 0.98 | 0.98 | 0.99 | 0.99 | 0.02 | 0.01 | 0.01 | 0.00 | 0.00 | 0.00 | 0.00 |
| Namib–2 | 0.74 | 0.82 | 0.86 | 0.92 | 0.94 | 0.96 | 0.97 | 0.03 | 0.02 | 0.02 | 0.01 | 0.01 | 0.01 | 0.01 |
| **Southern Africa (mean and st. dev.)** | **0.83** | **0.88** | **0.91** | **0.95** | **0.96** | **0.98** | **0.98** | **0.12** | **0.09** | **0.07** | **0.04** | **0.03** | **0.02** | **0.02** |
| Australia | 0.70 | 0.81 | 0.85 | 0.91 | 0.93 | 0.96 | 0.97 | 0.04 | 0.03 | 0.02 | 0.01 | 0.01 | 0.01 | 0.01 |
| **Australia (mean and st. dev.)** | **0.70** | **0.81** | **0.85** | **0.91** | **0.93** | **0.96** | **0.97** | **0.04** | **0.03** | **0.02** | **0.01** | **0.01** | **0.01** | **0.01** |
| | | | | | | | | | | | | | | |
| **Full dataset (mean and st. dev.)** | **0.85** | **0.91** | **0.93** | **0.96** | **0.96** | **0.97** | **0.98** | **0.08** | **0.05** | **0.04** | **0.03** | **0.02** | **0.01** | **0.01** |
| **Full dataset median** | **0.88** | **0.92** | **0.94** | **0.96** | **0.97** | **0.98** | **0.98** | | | | | | | |
| **Full dataset 10% percentile** | **0.74** | **0.84** | **0.88** | **0.92** | **0.94** | **0.96** | **0.96** | | | | | | | |
| **Full dataset 90% percentile** | **0.93** | **0.96** | **0.97** | **0.98** | **0.99** | **0.99** | **0.99** | | | | | | | |

1226

Table 6. Results of the linear fit between $k$ and SSA and the mass concentration of iron oxides, $MC_{Fe-ox\%}$, hematite, $MC_{Hem\%}$, goethite, $MC_{Goeth\%}$, and elemental iron, $MC_{Fe\%}$ in dust. Column 1 indicates the wavelength; $(a \pm \sigma a)$ indicates the retrieved slope and its estimated uncertainty; $(b \pm \sigma b)$ indicates the retrieved intercept and its estimated uncertainty; $R^2$ denotes the correlation coefficient and $\chi^2_{red}$ is the reduced chi–square of the fit.

| | $k = a\,MC_{Fe-ox\%} + b$ | | | $SSA = a\,MC_{Fe-ox\%} + b$ | | |
|---|---|---|---|---|---|---|
| Wavelength (nm) | $a \pm \sigma a$ | $b \pm \sigma b$ | $R^2$ ; $\chi^2_{red}$ | $a \pm \sigma a$ | $b \pm \sigma b$ | $R^2$ ; $\chi^2_{red}$ |
| 370 | $(11.9 \pm 2.4)\,10^{-4}$ | $(2.4 \pm 4.6)\,10^{-4}$ | 0.88 ; 0.6 | $(-5.8 \pm 0.8)\,10^{-2}$ | $(1.00 \pm 0.02)$ | 0.83 ; 1.7 |
| 470 | $(9.0 \pm 1.7)\,10^{-4}$ | $(1.7 \pm 3.2)\,10^{-4}$ | 0.89 ; 0.8 | $(-3.8 \pm 0.6)\,10^{-2}$ | $(1.00 \pm 0.01)$ | 0.78 ; 1.8 |
| 520 | $(6.8 \pm 1.3)\,10^{-4}$ | $(1.3 \pm 2.4)\,10^{-4}$ | 0.90 ; 0.9 | $(-2.9 \pm 0.4)\,10^{-2}$ | $(1.01 \pm 0.01)$ | 0.76 ; 2.0 |
| 590 | $(4.5 \pm 0.9)\,10^{-4}$ | $(0.9 \pm 1.6)\,10^{-4}$ | 0.85 ; 1.4 | $(-1.8 \pm 0.3)\,10^{-2}$ | $(1.00 \pm 0.01)$ | 0.75 ; 2.3 |
| 660 | $(4.3 \pm 0.8)\,10^{-4}$ | $(0.8 \pm 1.4)\,10^{-4}$ | 0.81 ; 1.6 | $(-1.3 \pm 0.2)\,10^{-2}$ | $(1.00 \pm 0.00)$ | 0.75 ; 2.2 |
| 880 | $(3.4 \pm 0.6)\,10^{-4}$ | $(0.6 \pm 1.2)\,10^{-4}$ | 0.79 ; 1.0 | $(-0.76 \pm 0.16)\,10^{-2}$ | $(1.00 \pm 0.00)$ | 0.79 ; 1.4 |
| 950 | $(3.2 \pm 0.6)\,10^{-4}$ | $(0.6 \pm 1.0)\,10^{-4}$ | 0.77 ; 1.1 | $(-0.62 \pm 0.13)\,10^{-2}$ | $(0.99 \pm 0.00)$ | 0.78 ; 1.1 |

| | $k = a\,MC_{Hem\%} + b$ | | | $SSA = a\,MC_{Hem\%} + b$ | | |
|---|---|---|---|---|---|---|
| Wavelength (nm) | $a \pm \sigma a$ | $b \pm \sigma b$ | $R^2$ ; $\chi^2_{red}$ | $a \pm \sigma a$ | $b \pm \sigma b$ | $R^2$ ; $\chi^2_{red}$ |
| 370 | $(9.7 \pm 2.7)\,10^{-4}$ | $(2.7 \pm 4.0)\,10^{-4}$ | 0.67 ; 1.9 | $(-4.4 \pm 0.6)\,10^{-2}$ | $(0.95 \pm 0.01)$ | 0.73 ; 3.5 |
| 470 | $(8.3 \pm 1.9)\,10^{-4}$ | $(1.9 \pm 2.7)\,10^{-4}$ | 0.72 ; 1.9 | $(-3.0 \pm 0.4)\,10^{-2}$ | $(0.97 \pm 0.01)$ | 0.76 ; 3.2 |
| 520 | $(6.9 \pm 1.5)\,10^{-4}$ | $(1.5 \pm 2.0)\,10^{-4}$ | 0.74 ; 2.0 | $(-2.2 \pm 0.3)\,10^{-2}$ | $(0.98 \pm 0.00)$ | 0.78 ; 3.3 |
| 590 | $(3.7 \pm 0.8)\,10^{-4}$ | $(0.9 \pm 1.2)\,10^{-4}$ | 0.61 ; 2.1 | $(-1.3 \pm 0.2)\,10^{-2}$ | $(0.99 \pm 0.00)$ | 0.71 ; 2.7 |
| 660 | $(3.7 \pm 0.8)\,10^{-4}$ | $(0.8 \pm 1.1)\,10^{-4}$ | 0.51 ; 2.6 | $(-0.9 \pm 0.2)\,10^{-2}$ | $(0.99 \pm 0.00)$ | 0.62 ; 2.5 |
| 880 | $(2.9 \pm 0.7)\,10^{-4}$ | $(0.7 \pm 1.1)\,10^{-4}$ | 0.43 ; 2.1 | $(-0.6 \pm 0.1)\,10^{-2}$ | $(0.99 \pm 0.00)$ | 0.57 ; 1.8 |
| 950 | $(2.6 \pm 0.6)\,10^{-4}$ | $(0.6 \pm 0.9)\,10^{-4}$ | 0.46 ; 2.1 | $(-0.5 \pm 0.1)\,10^{-2}$ | $(0.99 \pm 0.00)$ | 0.49 ; 1.7 |

| | $k = a\,MC_{Goeth\%} + b$ | | | $SSA = a\,MC_{Goeth\%} + b$ | | |
|---|---|---|---|---|---|---|
| Wavelength (nm) | $a \pm \sigma a$ | $b \pm \sigma b$ | $R^2$ ; $\chi^2_{red}$ | $a \pm \sigma a$ | $b \pm \sigma b$ | $R^2$ ; $\chi^2_{red}$ |
| 370 | $(9.0 \pm 2.5)\,10^{-4}$ | $(2.5 \pm 2.2)\,10^{-4}$ | 0.47 ; 1.8 | $(-13.4 \pm 6.9)\,10^{-3}$ | $(0.90 \pm 0.01)$ | 0.32 ; 6.8 |
| 470 | $(5.5 \pm 1.7)\,10^{-4}$ | $(1.7 \pm 1.5)\,10^{-4}$ | 0.43 ; 2.3 | $(-8.3 \pm 4.7)\,10^{-3}$ | $(0.94 \pm 0.00)$ | 0.21 ; 6.2 |
| 520 | $(3.4 \pm 1.1)\,10^{-4}$ | $(1.1 \pm 1.2)\,10^{-4}$ | 0.41 ; 2.5 | $(-4.9 \pm 3.2)\,10^{-3}$ | $(0.96 \pm 0.00)$ | 0.17 ; 6.4 |
| 590 | $(0.5 \pm 0.6)\,10^{-4}$ | $(0.6 \pm 0.8)\,10^{-4}$ | 0.50 ; 3.2 | $(0.9 \pm 2.0)\,10^{-3}$ | $(0.97 \pm 0.00)$ | 0.23 ; 5.5 |
| 660 | $(2.2 \pm 0.8)\,10^{-4}$ | $(0.8 \pm 0.7)\,10^{-4}$ | 0.55 ; 3.6 | $(0.2 \pm 1.6)\,10^{-3}$ | $(0.98 \pm 0.00)$ | 0.34 ; 4.4 |
| 880 | $(2.6 \pm 0.8)\,10^{-4}$ | $(0.8 \pm 0.6)\,10^{-4}$ | 0.62 ; 2.4 | $(-1.1 \pm 1.4)\,10^{-3}$ | $(0.98 \pm 0.00)$ | 0.47 ; 3.0 |
| 950 | $(2.6 \pm 0.8)\,10^{-4}$ | $(0.8 \pm 0.6)\,10^{-4}$ | 0.55 ; 2.5 | $(-2.1 \pm 1.4)\,10^{-3}$ | $(0.98 \pm 0.00)$ | 0.54 ; 2.6 |

| | $k = a\,MC_{Fe\%} + b$ | | | $SSA = a\,MC_{Fe\%} + b$ | | |
|---|---|---|---|---|---|---|
| Wavelength (nm) | $a \pm \sigma a$ | $b \pm \sigma b$ | $R^2$ ; $\chi^2_{red}$ | $a \pm \sigma a$ | $b \pm \sigma b$ | $R^2$ ; $\chi^2_{red}$ |
| 370 | $(6.0 \pm 1.4)\,10^{-4}$ | $(1.4 \pm 0.7)\,10^{-4}$ | 0.60 ; 1.5 | $(-2.7 \pm 0.4)\,10^{-2}$ | $(1.02 \pm 0.02)$ | 0.67 ; 3.1 |
| 470 | $(4.7 \pm 1.0)\,10^{-4}$ | $(1.0 \pm 0.5)\,10^{-4}$ | 0.62 ; 1.7 | $(-1.8 \pm 0.3)\,10^{-2}$ | $(1.02 \pm 0.01)$ | 0.72 ; 2.8 |
| 520 | $(3.9 \pm 0.8)\,10^{-4}$ | $(0.8 \pm 3.9)\,10^{-4}$ | 0.65 ; 1.6 | $(-1.3 \pm 0.2)\,10^{-2}$ | $(1.01 \pm 0.01)$ | 0.72 ; 2.9 |
| 590 | $(2.5 \pm 0.5)\,10^{-4}$ | $(0.5 \pm 2.4)\,10^{-4}$ | 0.56 ; 1.7 | $(-0.8 \pm 0.1)\,10^{-2}$ | $(1.01 \pm 0.01)$ | 0.70 ; 2.4 |
| 660 | $(2.0 \pm 0.4)\,10^{-4}$ | $(0.4 \pm 1.7)\,10^{-4}$ | 0.48 ; 1.9 | $(-0.5 \pm 0.1)\,10^{-2}$ | $(1.00 \pm 0.00)$ | 0.62 ; 2.0 |
| 880 | $(1.8 \pm 0.4)\,10^{-4}$ | $(0.4 \pm 2.0)\,10^{-4}$ | 0.40 ; 1.8 | $(-0.4 \pm 0.1)\,10^{-2}$ | $(1.00 \pm 0.00)$ | 0.54 ; 1.6 |
| 950 | $(1.4 \pm 0.3)\,10^{-4}$ | $(0.3 \pm 1.4)\,10^{-4}$ | 0.45 ; 2.0 | $(-0.3 \pm 0.1)\,10^{-2}$ | $(1.00 \pm 0.00)$ | 0.49 ; 1.5 |

**Figure 1.** Flowchart illustrating the procedure for data treatment and retrieval of physical and chemical
(size, composition) and spectral optical properties (single scattering albedo, SSA, and complex refrac-
tive index) of mineral dust aerosols. In red we mention the different corrections performed and the
values adopted in the calculations.

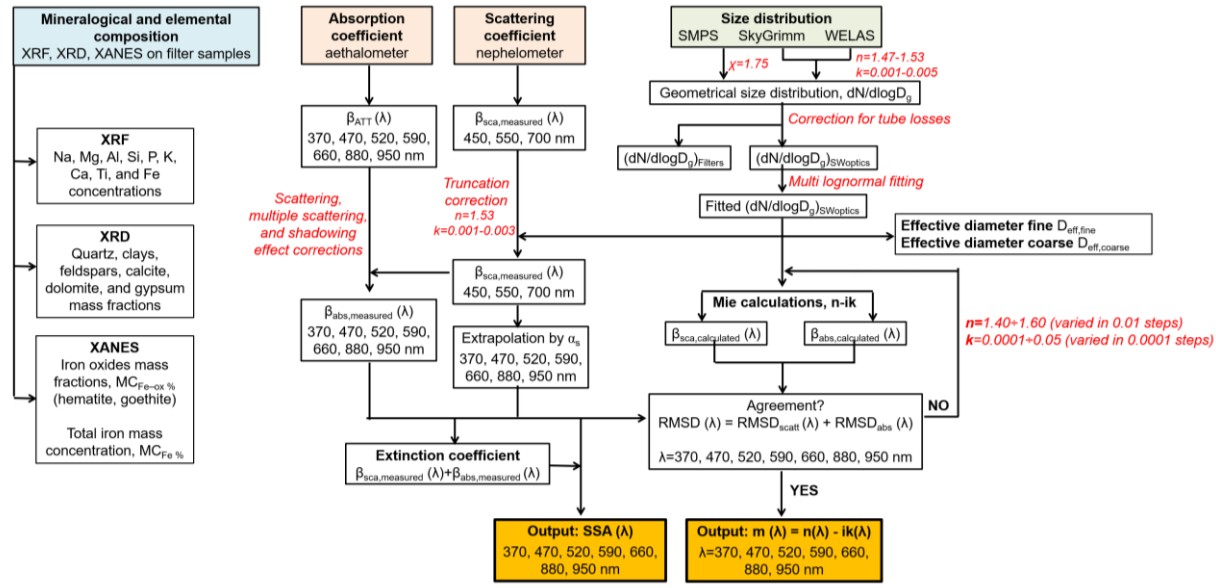


**Figure 2.** Box and whisker plot showing the full variability of hematite and goethite mass fractions in
the soils for the clay–sized (<2 μm diameter) and silt–sized (<60 μm diameter) fractions as retrieved
from the global soil mineralogical database by Journet et al. (2014). The box and whisker plot include
data for the nine desert source areas depicted in Ginoux et al. (2012) and DB17 (Northern Africa, Sahel,
Eastern Africa and the Middle East, Central Asia, Eastern Asia, North America, South America, South-
ern Africa, and Australia). Dots indicate hematite and goethite content in clay–sized and silt–sized soils
(always from Journet et al.) extracted in correspondence to the geographical coordinates where the
nineteen soils used in the CESAM experiments were collected. The Journet et al. database assumes
that the iron oxides in the silt fraction consist only of goethite.

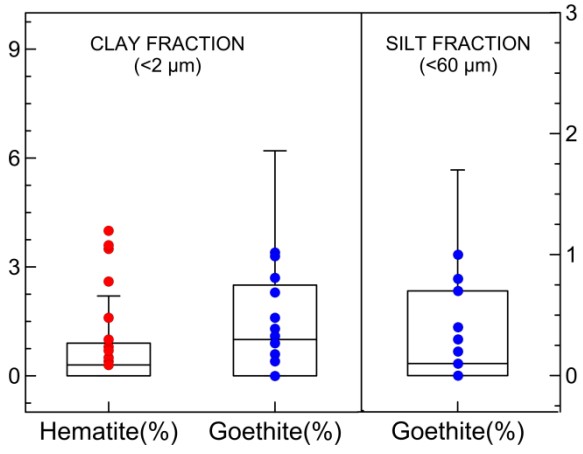




















**Figure 3.** Top panel: time series of the aerosol mass concentration (cross symbols) and effective fine ($D_{eff,fine}$, open dots) and coarse diameter ($D_{eff,coarse}$, open squares) measured inside the CESAM chamber (red symbols) and at the input of the SW instruments (black symbols) for one experiment (Morocco dust). Bottom panel: time series of the scattering $\beta_{sca}$ and absorption $\beta_{abs}$ coefficients at 370 nm for the same experiment. Mass concentrations are reported as 6–sec data, while all other quantities are 10–min averages.

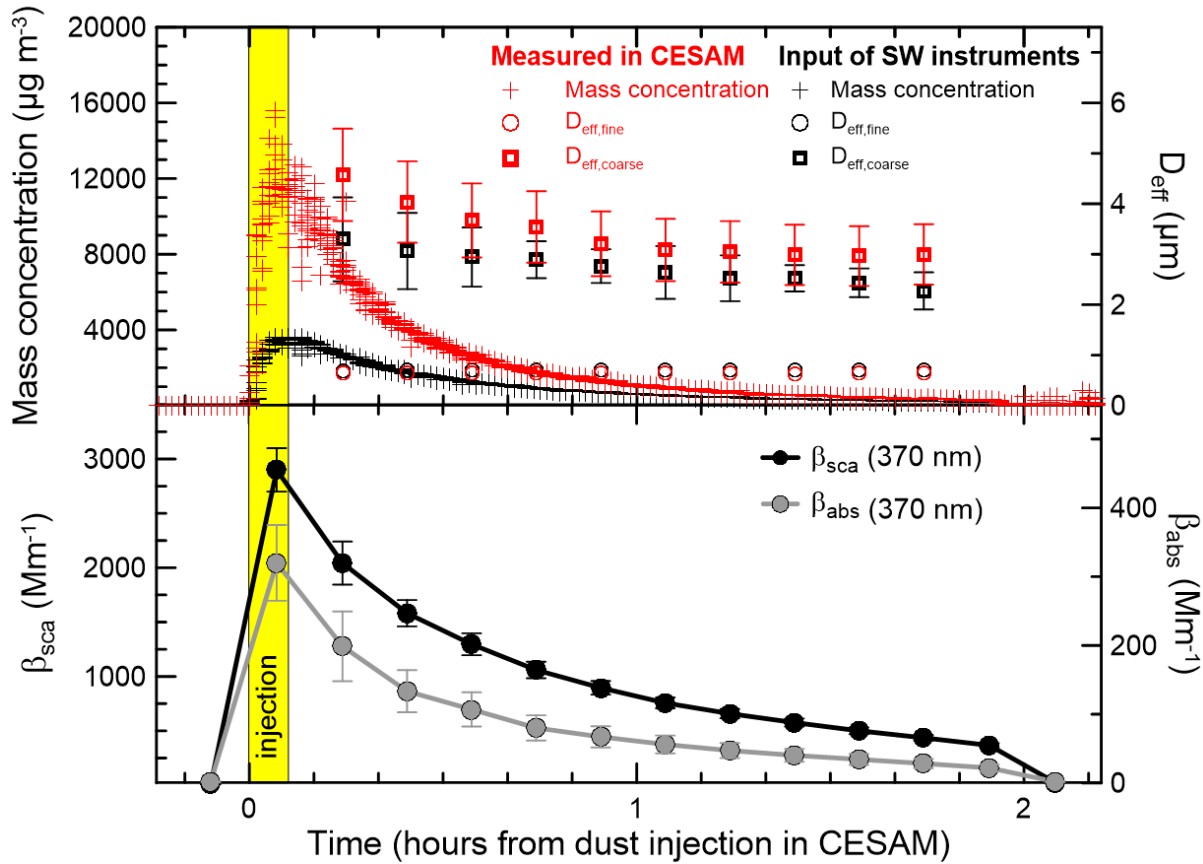

**Figure 4.** Comparison of dust size distributions sensed by the SW optical instruments (behind the SW instruments inlet $(dV/dlogD_g)_{SWoptics}$), with field data for long–range transported dust. The thick black line represents the mean value of $(dV/dlogD_g)_{SWoptics}$ at the peak of the dust injection in CESAM for experiments with the different samples. The grey shaded area indicates the range of $(dV/dlogD_g)_{SWoptics}$ for all samples. The dotted black line shows the average of the dust size distribution at the peak of the injection inside the CESAM chamber from DB17. Field data are from: Formenti et al. (2001) (CLAIRE campaign in Suriname, South America), Maring et al. (2003) and Denjean et al. (2016b) (PRIDE and DUST–ATTACK campaigns in Puerto Rico, Caraibes), Müller et al. (2011), Chen et al. (2011) and Ryder et al. (2018) (SAMUM2, NAMMA, and AER–D campaigns in Cape Verde, eastern Atlantic), and Weinzierl et al. (2017) (SALTRACE campaign, data from Barbados). For comparison, data taken close to the source in Niger from Formenti et al. (2011) during the AMMA campaign are also shown. SAL stands for Saharan Air Layer. All data are reported as volume size distributions normalised at the maximum.
*(The different acronyms spell out as: AER–D= AERosol Properties – Dust; AMMA = African Monsoon Multidisciplinary Analysis; CLARE= Cooperative LBA Airborne Regional Experiment; DUST–ATTACK+ Dust Aging and TransporT from Africa to the Caribbean; NAMMA = NASA African Monsoon Multidisciplinary Analysis; PRIDE = Puerto Rico Dust Experiment; SALTRACE= Saharan Aerosol Long–range Transport and Aerosol–Cloud–Interaction Experiment; SAMUM = Saharan Mineral Dust Experiment).*

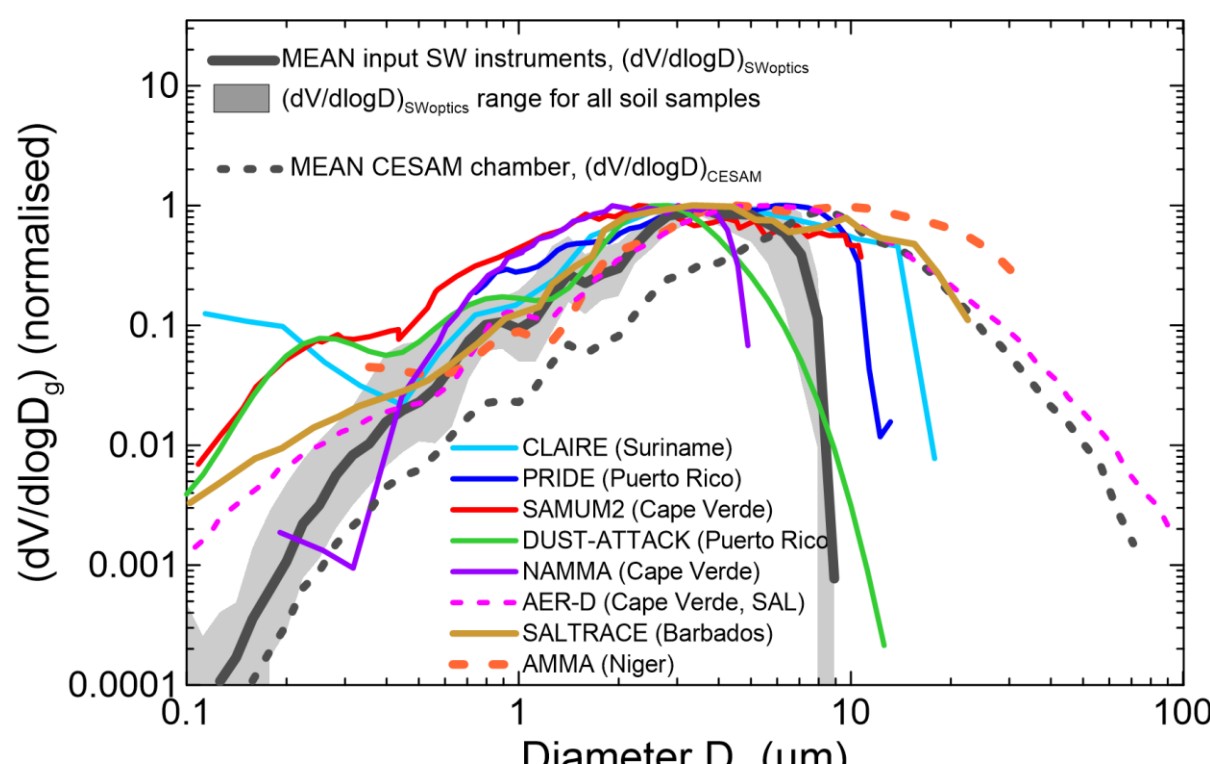

**Figure 5.** Spectral extinction coefficient, absorption coefficient, SSA, and real (*n*) and imaginary (*k*) parts of the refractive index at the peak of the dust injection in the chamber and after 30 and 90 minutes for Morocco and Algeria dust samples. Data are reported at the seven aethalometer wavelengths (370, 470, 520, 590, 660, 880, and 950 nm) as 10–min averages. In the top panel we report the extinction calculated as the sum of scattering and absorption coefficients. For the sake of clarity error bars are not shown for SSA, *n*, and *k* data.

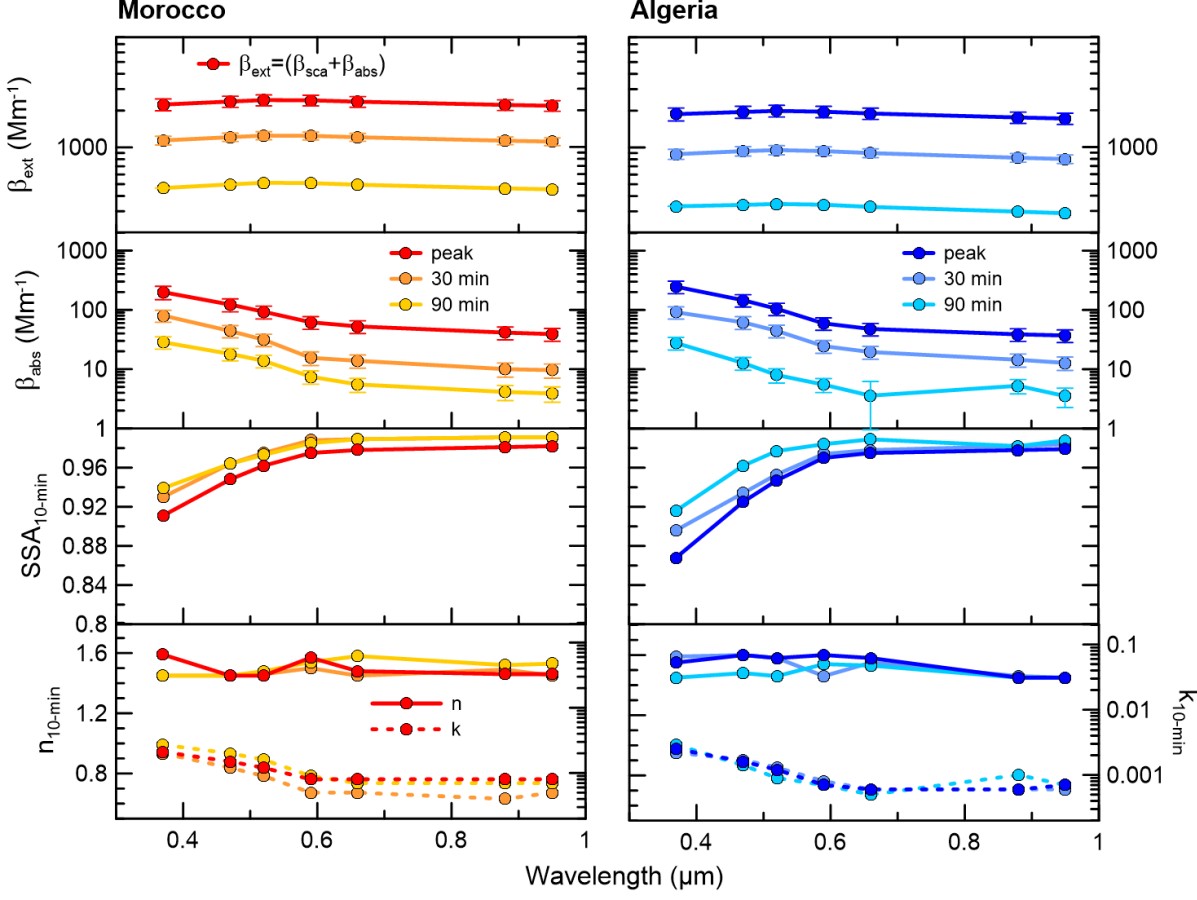

**Figure 6.** Real (*n*) and imaginary (*k*) parts of the dust complex refractive index at seven wavelengths
between 370 and 950 nm obtained for the 19 aerosol samples analyzed in this study. Data correspond
to the time average of the 10 min values obtained between the peak of the injection and 120 min later.
The error bar corresponds to the absolute uncertainty in *n* and *k*, estimated to be <8% for n and between
13 and 75 % for *k*.

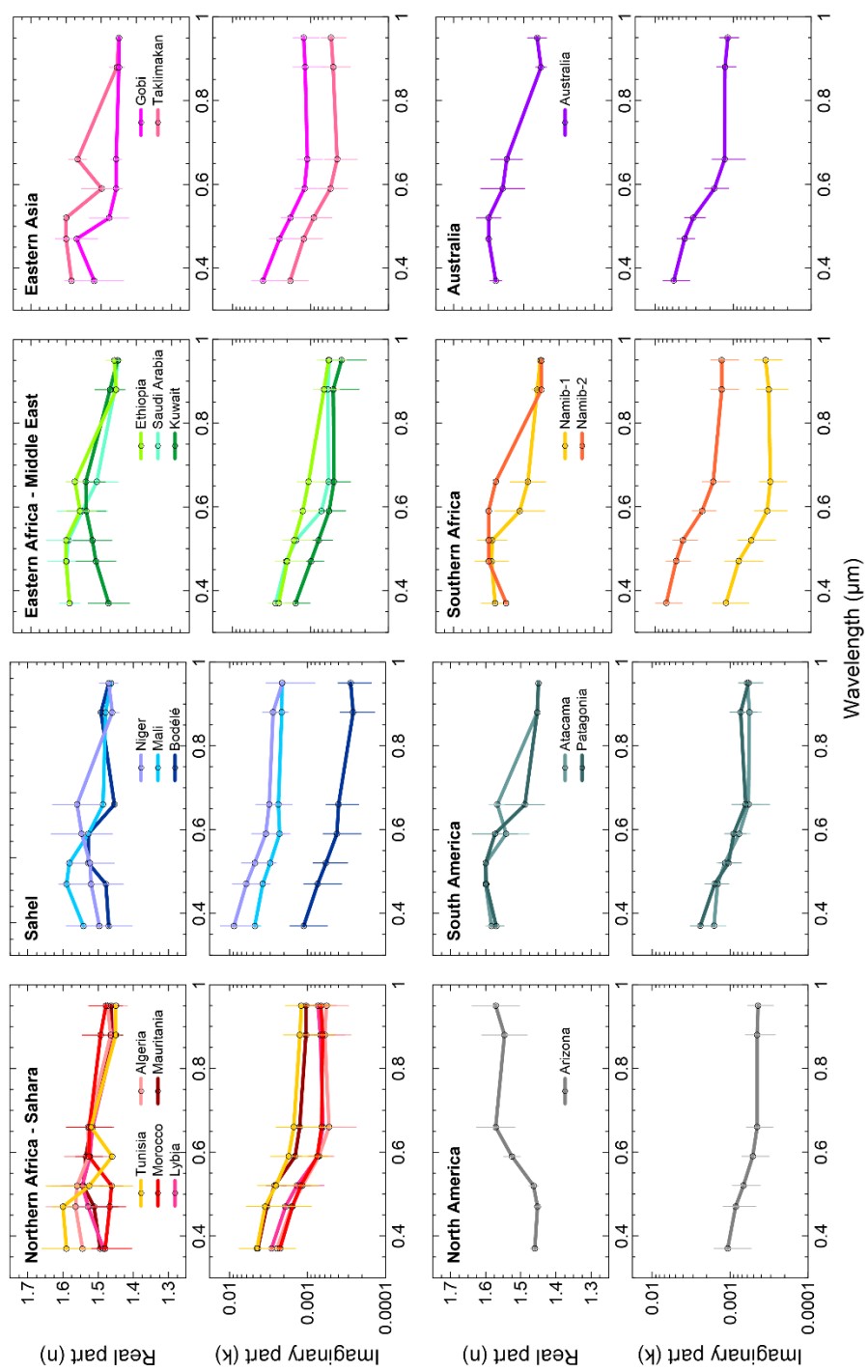


**Figure 7.** Single scattering albedo (SSA) at seven wavelengths between 370 and 950 nm obtained for
the 19 aerosol samples analyzed in this study. Data correspond for each sample (with the exception of
Tunisia and Namib–2, see Sect. 3.1) to the fit of the 10 min values of $\beta_{sca}$ versus $\beta_{abs}$, and the uncertainty
is between 1% and 12% at 370 nm and between 1% and 3% at 950 nm.

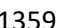






**Figure 8.** Comparison of results obtained in this study with literature–compiled values of the (a) dust real and imaginary parts of the refractive index (*n, k*) and (b) single scattering albedo (SSA) in the SW spectral range. The regions in grey indicate the full range of variability obtained in this study, and the black thick lines are the means of *n*, *k* and SSA obtained for the different aerosol samples. Literature values include estimates from ground–based and aircraft observations during field campaigns, laboratory studies, AERONET inversions, and estimates from dust mineralogical composition. Data are in some cases for the full dust size distribution, while in other only the fine fraction below about 2 µm is represented (identified with *).

The main provenance of the dust and datasets from the literature is provided in the following: Volz et al. (1972) is data for rainout dust collected in Germany; Patterson et al. (1977) is Saharan dust; Hess et al. (1998) is data from the OPAC database; Colarco et al. (2002) is data for dust from Dakar, Sal, and Tenerife; Dubovik et al. (2002) included data from Bahrain–Persian Gulf and Solar Village–Saudi Arabia AERONET stations; Haywood et al. (2003) is dust from Mauritania; Sinyuk et al. (2003) is data from Cape Verde, Dakar, and Burkina Faso; Clarke et al. (2004) is Asian dust offshore of China, Japan, and Korea; Linke et al. (2006)–A is dust from Cairo; Linke et al. (2006)–B is dust from Morocco; Balkanski et al (2007) is calculated from mineralogical composition assuming a 1.5% hematite mass fraction in dust; Todd et al. (2007) is from Bodélé; Osborne et al. (2008) is from Niger; Otto et al. (2009), Petzold et al. (2009), Schladitz et al. (2009), and Muller et al. (2010, 2011) is dust originated mostly in Morocco; McConnell et al. (2008, 2010) is dust from Niger/Senegal; Chen et al. (2011) is dust from Western Sahara; Formenti et al. (2011) in the *k* plot is an average of airborne observations for the AMMA campaign in Niger, while for the SSA plot, Formenti et al. (2011)–A is from observations in the Saharan Air Layer, –B is from Bodélé/Sudan, and –C is a Sahelian uplift episode; Johnson et al. (2011) is dust from Western Sahara; Moosmüller et al. (2012) analysed samples from Middle East, Mali and Spain, and here we report the average of their obtained values; Wagner et al. (2012) obtained *k* values for several samples from Burkina Faso, Cairo and the SAMUM campaign and here we report the values for the maximum of their spectral *k* (Burkina Faso) and the minimum (Cairo); Ryder et al. (2013) is dust from Western Sahara and Mauritania and we report in both *k* and SSA plots the average of their observations; Engelbrecht et al. (2016) analysed many dust samples from all over the world, here we report their estimated minimum and maximum of the dust SSA that are –A from California and –B from the Etosha Pan in Namibia; Stegmann and Yang (2017) modelled the refractive index of dust based on assumed mineralogical compositions typical for Northern and Southern Sahara and Western and Eastern Asia dust, and here we report the average of their results for both *n* and *k*. Uncertainties in the field observations have been omitted for the sake of clarity. The legend identifies the line styles used in the plots.

*(The different acronyms spell out as (see also the caption of Fig. 4): AERONET = Aerosol Robotic Network; OPAC = Optical Properties of Aerosols and Clouds; SHADE = Saharan Dust Experiment; BODEX = The Bodélé Dust Experiment; DABEX = Dust and Biomass Experiment; SAMUM1 and SAMUM2 refers to the two SAMUM campaigns in Morocco and Cape Verde, respectively, SAMUM = Saharan Mineral Dust Experiment; DODO = Dust Outflow and Deposition to the Ocean; ACE–Asia =*

 *Asian Pacific Regional Aerosol Characterization Experiment; GERBILS = Geostationary Earth Radia-*

*tion Budget Intercomparison of Longwave and Shortwave radiation).*


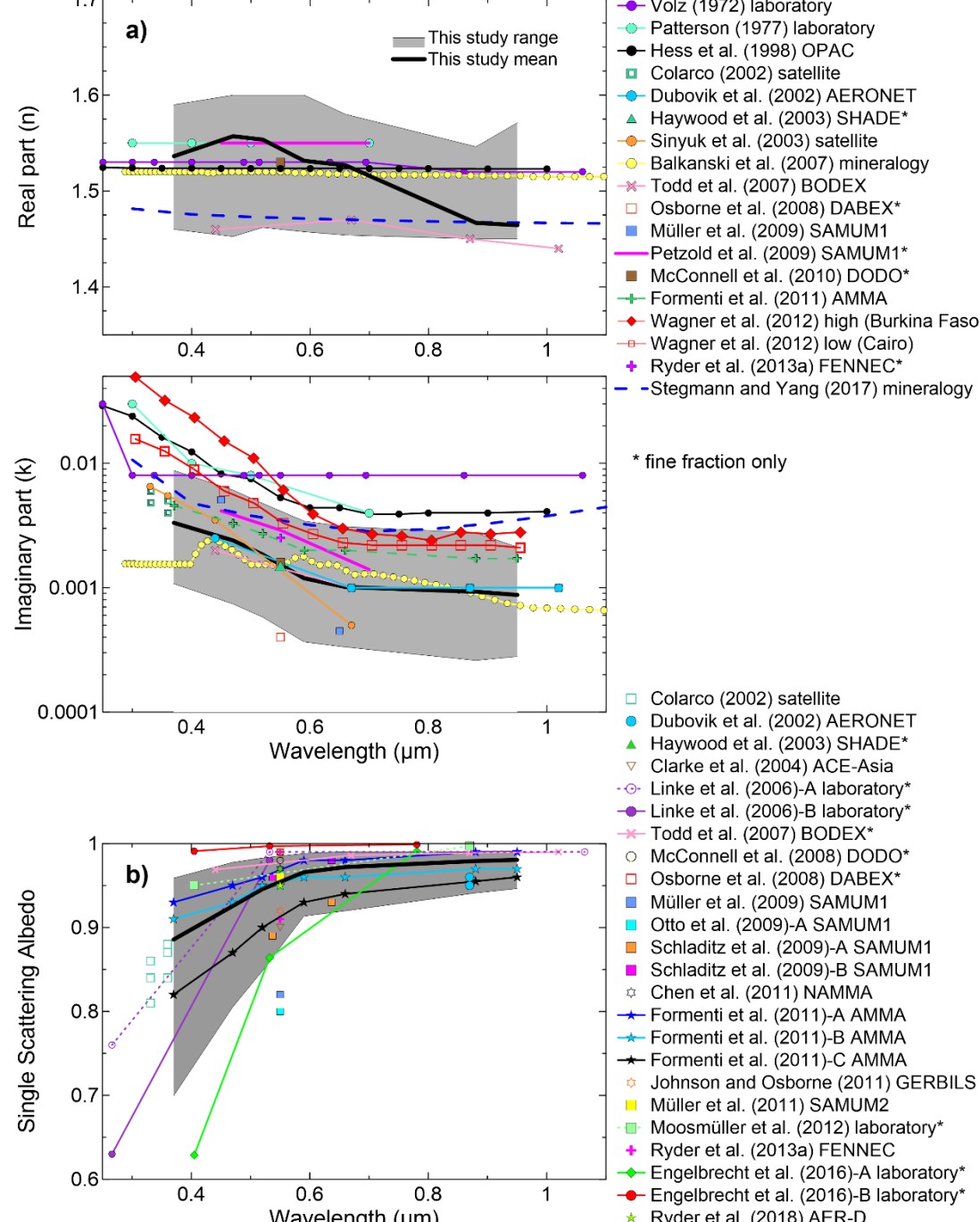








**Figure 9.** Experiment–averaged imaginary part of the refractive index ($k$, top panels) and single scattering albedo (SSA, bottom panels) at 370, 520, and 950 nm versus the mass concentration of iron oxides ($MC_{Fe–ox\%}$), hematite ($MC_{Hem\%}$), goethite ($MC_{Goeth\%}$), and elemental iron ($MC_{Fe\%}$) measured for the different dust samples analysed in this study. The calculated linear fit regression lines are shown, together with the correlation coefficients of the fits ($R^2$). The legend indicates the line styles used in the plots. Data for the Taklimakan sample were excluded from the $k$ and SSA plots versus $MC_{Fe–ox\%}$, $MC_{Hem\%}$, and $MC_{Goeth\%}$ due to the absence of data for this sample.

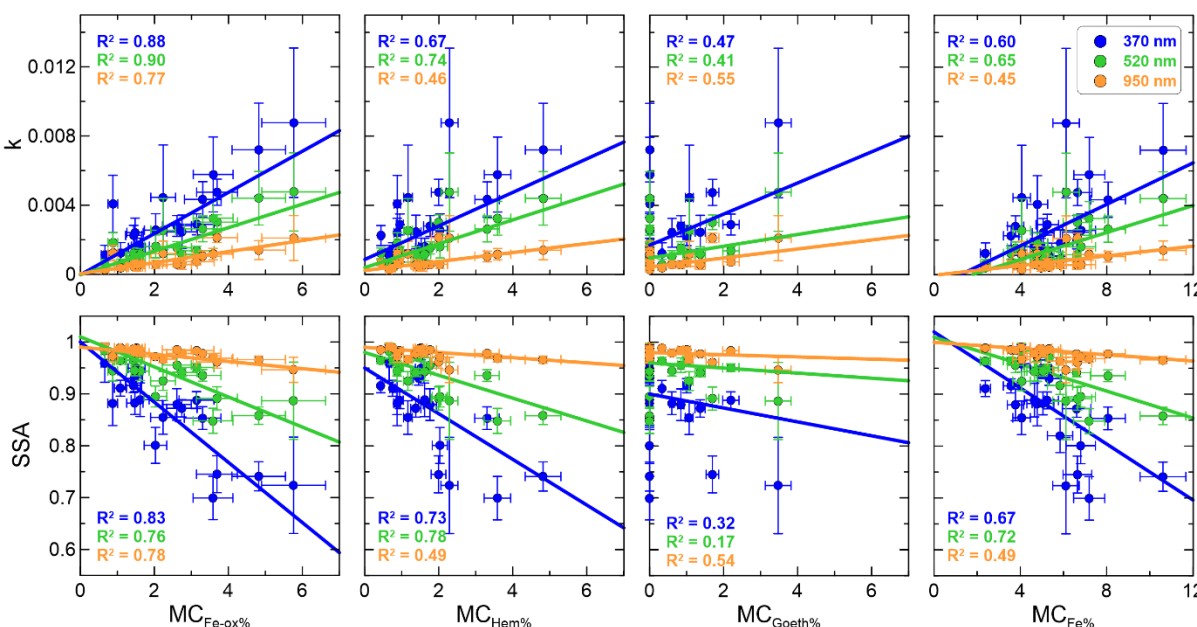

**Figure 10.** 10–min averaged imaginary refractive index ($k_{10-min}$, top panels) and single scattering albedo
(SSA$_{10-min}$, bottom panels) at 370, 520, and 950 nm versus effective coarse diameter ($D_{eff,coarse}$) esti-
mated at the input of the SW instruments. Data were classified in three classes based on the iron oxide
content of the dust samples. The linear fit curves and the correlation coefficients for the linear regression
fits for each dataset are also reported. The legend identifies the line styles used in the plots.


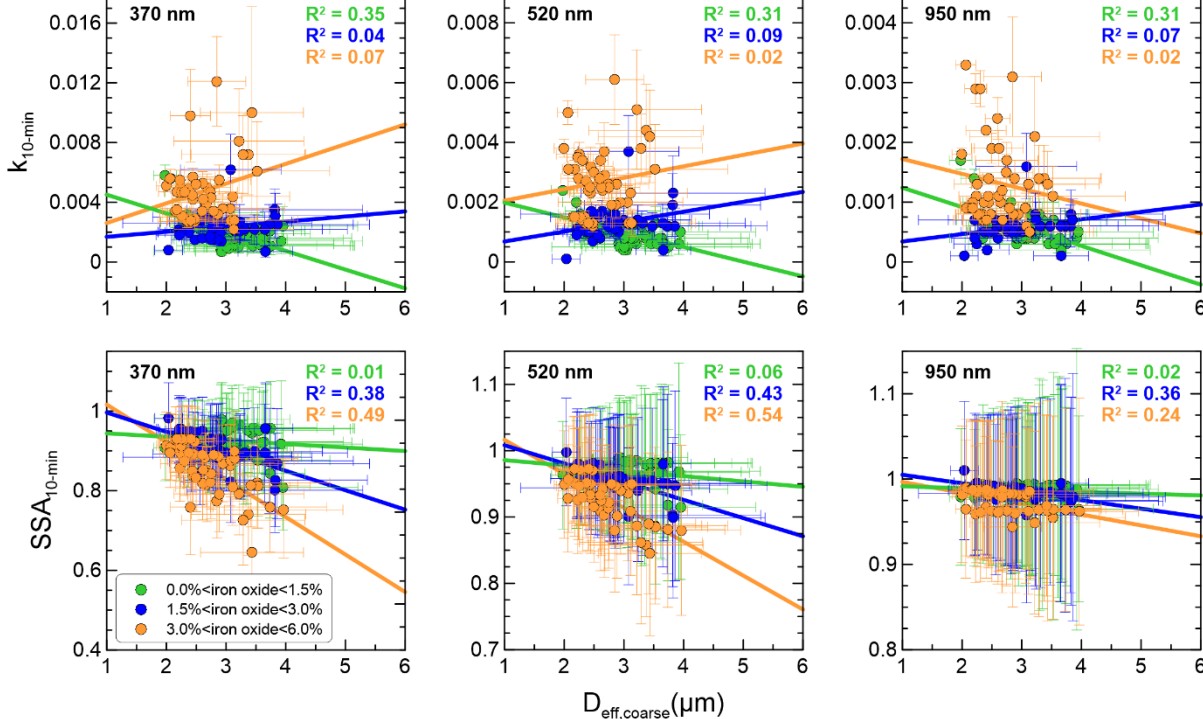
