# Peer review of "Complex refractive indices and single scattering albedo of global dust aerosols in the shortwave spectrum and relationship to size and iron content"

_Atmospheric Chemistry and Physics, 2019_

## Referee Comment (RC1) · Claire Ryder (Referee) · 15 Apr 2019

Overall comment

This paper provides new data on the spectral shortwave refractive index of mineral dust. Different global dust samples are suspended inside a chamber, and subsequently scattering, absorption, size distribution and composition are measured, and the former are used to calculate the refractive index. A relationship between iron oxide content and refractive index is also found. The authors find that the imaginary (absorption)

[Figure]

component is relatively low compared to the range of older datasets.

There is a serious lack of high quality dust refractive index data available in the literature, so this research is valuable and welcomed. The results will be important in providing refractive index data for any sort of atmospheric model representing dust (NWP, GCMs), as well as satellite retrieval algorithms, and I would expect the data from this paper to be widely used. The paper is of a high standard, clearly setting out methodology, propagation of uncertainties, and results. Several of the figures are very small and need to be made larger, but other than this I suggest only minor clarifications and suggestions.

General Comments

Do similar chamber studies exist, such that estimates/measurements of refractive index have been attempted in a chamber? (perhaps not, other than the preceding studies already mentioned done by the same group). In the case that they do, the authors should mention them in the introduction, if relevant. If not, I suggest the authors point this out, as it increases the novelty and originality of the work done for this paper.

In the abstract and conclusion, please add information on optical properties/refractive index (RI) at either 520 or 590nm. $\sim$550nm is the wavelength most frequently required for these properties, and also represents the peak solar intensity, so this wavelength (or one close to it) would be most useful to state properties at, in addition to 370 and 950nm already provided.

Specific Comments

Abstract - I suggest adding a sentence reflecting how the new RI results compare to older datasets – i.e. the real part is similar but the imaginary part falls at the low end of the published range. This is an important finding.

P2 l46 – 'an intrinsic property of matter' – although this is true, I suggest rewording, such as, 'k was found to be independent of size' since several other studies have found

k to be size-dependent.

L85 – insert 'geographic' before 'differences persist. . .'

L106-108 – It would be worth saying why models still assume the same dust composition globally – e.g. due to computational cost of additional tracers? And/or lack of a globally consistent information dataset?

L207 – 'Ogreen' typo

L257 – RI range for n and k – which values were used? Different values for different experiments? Where/how is this range applied?

L268 – 'cut at 10 microns' - this contradicts p5 l178 which says 5 microns/8 microns (50/100% efficiency).

L278-279- It would be useful to mention the cut-off diameters again here as deff,coarse does not represent the full size range.

L280 – it would be useful to say why modes were fitted to the size distribution, for non-experts.

L283 – how were modes fitted? (I think this is provided later in the paper but it should be mentioned here).

L301 – what about other uncertainties to the size distribution, such as shape assumptions and/or Mie-regime singularities?

L342-3 – and also when comparing the RI data?

L402 – 'contrasting' → 'contradictory'?

L447 – Muller et al (2011) report observations at Capo Verde, not transported across the Atlantic.

L460-464 – And also the fact that the size distribution above the 50% transmission efficiency (5 microns?) is not well represented, should be mentioned.

[Figure]

Figure 4 and discussion in lines 451-464 – the authors should consider that some of the observational data they show from other campaigns was also restricted by maximum size measured or by inlet transmission efficiencies (e.g. NAMMA, PRIDE). Information on some of these restrictions are provided in Ryder et al. (2018), table 1. As such, some of these datasets likely underestimate the coarse mode size distribution. Transported dust size distributions are also available for the AER-D campaign in the same paper which would add to the data already shown in Figure 5 and did not suffer from inlet restrictions.

L466 – it would be useful to add a line on the importance of iron oxides vs elemental iron for the benefit of non-specialists.

L469 – 'Australia' – should this be Namib-2? (values are not consistent with those in the table).

L469 – 'Iron oxides account for 11 and 62% of the iron mass' – where do these values come from? They are not shown in table 3?

Section 4.1.2 – did iron content vary with particle size? Was that measured?

L487 – angstrom exponents across which wavelengths?

L549 – Steigmann – typo either here or in references

L527 – this is a very long paragraph and would benefit from being broken up a bit.

L557-560 – if this is the case regarding the Wagner dataset, wouldn't that make their dataset more reliable?

L544-560 – Comparing the new results to the older data is an interesting and important discussion point of the paper. It would be useful to expand this discussion to include a little more information on the methods of Wagner et al (2012) and Steigmann & Yang (2017) to allow the reader to better understand the different approaches. Are the authors able to justify that their method is more reliable than the other studies? This

is done for the comparison to the Volz & Patterson work, but not the other mentioned publications. It would help readers if the authors are able to justify their results more strongly, as this will enable readers of the paper to use the new data over the older datasets with a high level of confidence, rather than just adding to the spread of existing data.

L569-580 – can the authors comment on the fit for hematite vs goethite? Based on these results, is it therefore necessary to measure *both* hematite and goethite (rather than only hematite, as is sometimes done), in order to retrieve appropriate absorption estimates?

L589-593 – So is this several (or only one) data point per dust sample?

L596-598 – is the same true if you plot deff_coarse vs iron oxide content? Wouldn't this be a more direct comparison?

L599-601 'RI . . . is independent of size' – this is not always the case – e.g. Kandler et al. (2009) show that composition, and RI, change with size, and this is also reflected in the figures of Otto et al. (2009).

L604-605 – this was also found by Ryder et al. (2018) – i.e. SSA was dominantly dependent on composition.

Section 4.5 – do the authors have any thoughts on how much, if at all, their RI data may apply (or differ) for dust close to the source, when the coarse mode is more prevalent? Would it be worth mentioning this again in the conclusions as a potential area for future research?

Summary & concluding remarks – Do the authors have any suggestions for how to extrapolate RI at wavelengths lower than 370nm or larger than 950nm? These wavelengths are often required for spectral data within GCMs and/or NWP. Or if not, perhaps this should also be mentioned as an area requiring work in the future.

L625-627 – Can the authors comment on the reason for their k data falling at the lower

edge of the literature data, while SSA data seems very much in the middle of the literature range? This might be considered somewhat contradictory.

L646-648 – this sentence contradicts the discussion around figure 9 where the authors say that the coarse mode (deff) impacts the SSA – clearly the size distribution impacts the SSA too.

L657 – 'we propose. . . a set of regionally-averaged n, k and SSA values. . .' I wonder how useful this would really be – e.g. RI values are severely different for several samples in the same region – e.g. the Sahel, South Africa, East Asia (fig 6) so that even a regional representation may underestimate the sub-regional variability. Additionally there is the computational cost of additional tracers that may be required. The authors should mention some of these issues.

678 – 'in link' → linking?

Tables & Figures

Table 3 caption +/-10%, 15%, 10% - what exactly are these uncertainties? They seem very large for % values in the table of ∼1-10%?

Figure 2 caption – it's not quite clear which data is which – dots are Journet data, so are the box+whiskers Ginoux/DB17? Also, if the current work uses the same samples, shouldn't the iron oxide data be the same?

Figure 3 – x-axis label – 'time (hours)' – given this starts at @1330, is this time of day? Time from start of experiment would be more appropriate.

Figure 6 – these panels are all far too small to interpret. These should be made significantly larger. I also suggest changing the axis ranges so that the data takes up more than a minimal fraction of the plot.

Figure 7 caption – L1256 'measured' – I suggest using 'represented' instead as for several of these studies the properties shown (RI, SSA) are not directly measured, but

calculated from other more directly measured properties.

Figure 7 – These panels & data displayed are far too small to interpret. I suggest removing the left hand column (data from this study) since that is replicated on the right hand column and can be seen here. The literature data is virtually impossible to interpret. I suggest making all these plots much larger and also expanding the y-axis range for the k and n plots.

Figure 9 – This figure could also do with the plots being larger, and having more zoomed-in displays of the data.

[Figure]

---

## Referee Comment (RC2) · Carlos Pérez García-Pando (Referee) · 23 Jun 2019

The paper by Di Biagio et al. presents a new dataset of complex refractive indices and single scattering albedo (SSA) at 7 wavelengths in the shortwave range for dust aerosols generated from 19 soil samples collected from the 8 more prominent dust emission regions on Earth. The dataset was derived after resuspending dust into a smog chamber and measuring its spectral scattering and absorption coefficients and size distribution. The complex refractive index was estimated by Mie calculations combining the optical and size data, and the SSA was derived directly from the scattering

and absorption coefficients. In addition, bulk composition was measured for each sample to better understand the influence of mineralogy upon absorption.

A main result is that the retrieved imaginary part (k) is in the lower range of values currently found in the literature. The study also provides wavelength-dependent (strong) linear relationships between k and SSA, and the content of iron oxides that may help simplifying the implementation of the effect of dust mineralogy upon absorption in climate models.

This well-written paper provides a new and unprecedented dataset along with new insights on the dependencies of the absorption on the content of iron oxides. Therefore, I strongly recommend its publication after addressing some minor general and specific comments that can be found below.

General Comment

The authors make several assumptions in the optical calculations in order to retrieve k from the measured data. These assumptions are clearly summarized in section 3.3. First, the size distribution from the OPC and the scattering coefficient depend upon the refractive index; the approach was to set fixed values. Second, the OPC optical-to-geometrical diameter conversion and the nephelometer truncation correction (in addition to the retrieval algorithm in section 3.2) assume homogeneous spherical particles and therefore use Mie theory.

In the first case the authors calculate the sensitivity of the results to the assumption and demonstrate the robustness of their approach. However, the potentially large impact of the second assumption (shape) is not quantified. It is argued that shape was not measured and that improper assumptions on the particle shape or morphology may induce even larger errors. I find this last argument a little weak: 1) There are estimates of shape in the literature and recent studies (e.g. Kok et al., 2017) highlighting the potential importance of shape upon extinction (using triaxial spheroids); 2) the influence of shape upon the conversion from optical to geometrical diameter seems also quite

important (increasing the amount of fine particles compared to coarse ones; Huang and Kok, 2018); 3) The nephelometer truncation error should also be quite sensitive to shape; 4) The authors compare their results with other estimates in the literature; in page 15 they argue that the strong differences with Wagner et al. (2012) may be due to the choice of the optical theory (T-Matrix vs Mie), which somehow admits that assuming spherical particles may be a strong source uncertainty.

While I understand that accounting for shape is a complex issue beyond the scope of this paper, I nevertheless ask the authors to more clearly acknowledge this (potentially strong) uncertainty both in the abstract, discussion (section 3.3) and conclusions. I would also invite the authors to explore and report on this uncertainty in future studies.

Specific comments

Title: The current form of the title may induce to confusion (it is not clear whether size relates to iron size or dust size). I suggest two possibilities:

"Complex refractive indices and single scattering albedo of global dust aerosols in the shortwave spectrum and relationship to size and iron content"

or

"Complex refractive indices and single scattering albedo of global dust aerosols in the shortwave spectrum and relationship to iron content and dust size"

Abstract: Please emphasize that the linear relationship is stronger with iron oxides than with total iron. Also, indicate that your estimates are in the lower range of the literature values.

L197 to L217.: Please elaborate a bit on the assumption of spheres for the truncation correction and please connect it to an improved discussion in section 3.3 as suggested above.

L241: "varied between…. nm and…."

L268: 10 microns?

L301: what about shape?

L542: what do you mean by "our" population average of 1.8 % . . . .?

L557: Please develop more. This is quite important in my opinion as I highlighted above. Can you further discuss the differences with Wagner?

L600: k is an intrinsic property of matter by definition. Perhaps rephrase this sentence. k is independent of size if mineral composition does not change with size.

L642: I would rephrase this sentence. The variability is controlled by the iron oxide content. Using total iron partly reflects this because oxides are a large fraction of it but the linear relationship is obviously not as strong.

L650: This sentence is a little confusing. k is independent of size by definition. However, an "effective" k of a size distribution may change if mineralogy changes with size and size evolves with transport. In fact in L667 you further propose to investigate this potential dependence.

L657: the variability within regions can be very large though, please highlight that.

L675: you may mention and cite EMIT (Green et al., 2018) as a potential near-future source of high resolution surface mineralogy data for arid and semi-arid regions

Figures: Figure 1 is not readable, please increase the font size in the flowchart

References:

Green, R. O.; Mahowald, N. M.; Clark, R. N.; Ehlmann, B. L.; Ginoux, P. A.; Kalashnikova, O. V.; Miller, R. L.; Okin, G.; Painter, T. H.; Pérez García-Pando, C.; Realmuto, V. J.; Swayze, G. A.; Thompson, D. R.; Middleton, E.; Guanter, L.; Ben Dor, E.; Phillips, B. R. (2018) NASA's Earth Surface Mineral Dust Source Investigation. American Geophysical Union, Fall Meeting 2018, abstract #A24D-01

Huang, Y.; Kok, J. F. (2018) Harmonization of size distributions of dust at emission. American Geophysical Union, Fall Meeting 2018, abstract #A21I-2821

Kok, J.F., D.A. Ridley, Q. Zhou, R.L. Miller, C. Zhao, C.L. Heald, D.S. Ward, S. Albani, and K. Haustein, 2017: Smaller desert dust cooling effect estimated from analysis of dust size and abundance. Nature Geosci., 10, no. 4, 274-278, doi:10.1038/ngeo2912.
* * *

---

## Author Comment (AC1) · 11 Oct 2019

Revision of the paper

**Complex refractive indices and single scattering albedo of global dust aerosols in the shortwave spectrum and relationship to iron content and size**

Claudia Di Biagio[1], Paola Formenti[1], Yves Balkanski[2], Lorenzo Caponi[1,3], Mathieu Cazaunau[1], Edouard Pangui[1], Emilie Journet[1], Sophie Nowak[4], Meinrat O. Andreae[5,6], Konrad Kandler[7], Thuraya Saeed[8], Stuart Piketh[9], David Seibert[10], Earle Williams[11], and Jean–Francois Doussin[1]

The authors wish to thank the reviewers for their valuable comments which helped to improve the quality and readability of the manuscript. Answers to the reviewer's comments are reported in the following (questions in black, answers in red).

▪▪▪▪▪▪▪▪▪▪▪▪▪▪▪▪▪▪▪▪▪▪▪▪▪▪▪▪▪▪▪▪▪▪▪▪▪▪▪▪▪▪▪▪▪▪▪▪▪▪▪▪▪▪▪▪▪▪▪▪▪▪▪▪▪▪▪▪▪

We would like to draw attention on the fact that in Fig. 7 there was an error in the original dataset published by Stegmann and Yang (2017), which we corrected in this revised version (P. Stegmann, personal communication). Due to this change, part of the discussion in the text (Sect. 4.3) was modified since the new data from Stegmann and Yang (2017) now fit within the range of our dataset.

Also, based on the reviewer's comments, we revised all the different figures in the paper in order to improve their readability and also, where necessary, we modified the Figure's and Table's captions to include missing relevant information. We split Figure 6 in two, so we added a figure to the manuscript.

We additionally slightly revised the text in some sections to better improve its readability. In particular, we splitted the Summary and Concluding remarks in two separate sections.

We hope our changes will meet the reviewers' requirements and improve the quality of the paper.

All references cited in this document can be found in the main article.

**Claire Ryder (Referee #1)**

Overall comment

This paper provides new data on the spectral shortwave refractive index of mineral dust. Different global dust samples are suspended inside a chamber, and subsequently scattering, absorption, size distribution and composition are measured, and the former are used to calculate the refractive index. A relationship between iron oxide content and refractive index is also found. The authors find that the imaginary (absorption) component is relatively low compared to the range of older datasets. There is a serious lack of high quality dust refractive index data available in the literature, so this research is valuable and welcomed. The results will be important in providing refractive index data for any sort of atmospheric model representing dust (NWP, GCMs), as well as satellite retrieval algorithms, and I would expect the data from this paper to be widely used. The paper is of a high standard, clearly setting out methodology, propagation of uncertainties, and results. Several of the figures are very small and need to be made larger, but other than this I suggest only minor clarifications and suggestions.

General Comments

Do similar chamber studies exist, such that estimates/measurements of refractive index have been attempted in a chamber? (perhaps not, other than the preceding studies already mentioned done by the same group). In the case that they do, the authors should mention them in the introduction, if relevant. If not, I suggest the authors point this out, as it increases the novelty and originality of the work done for this paper.

As far as we know there exists only one other study that retrieved the spectral refractive index (only imaginary part in that case) for dust aerosols dispersed in a large chamber. This is the study by Wagner et al. (2012). We added the text below but not in the introduction as suggested by the reviewer but in the summary, to point out on the novelty of our work:

*"Some other laboratory studies were performed in the past to investigate the shortwave SSA of dust from different sources worldwide and its dependence on composition (Linke et al., 2006; Moosmüller et al., 2012; Engelbrecht et al., 2016). Conversely, for the refractive index it exists to our knowledge only one other chamber study, the one by Wagner et al. (2012), that retrieved the imaginary part k between 305 and 955 nm for dust aerosols from a limited number of source areas in Africa (Burkina Faso, Egypt and Morocco). As a matter of fact, our work provides the first consistent simulation chamber study of the complex refractive index of global dust."*

In the abstract and conclusion, please add information on optical properties/refractive index (RI) at either 520 or 590nm. 550nm is the wavelength most frequently required for these properties, and also represents the peak solar intensity, so this wavelength (or one close to it) would be most useful to state properties at, in addition to 370 and 950nm already provided.

As suggested by the reviewer we added information on optical properties at 520 nm both in the abstract and in the conclusions.

Specific Comments

Abstract – I suggest adding a sentence reflecting how the new RI results compare to older datasets – i.e. the real part is similar but the imaginary part falls at the low end of the published range. This is an important finding.

As suggested by the reviewer we added this comment to the abstract.

P2 l46 – 'an intrinsic property of matter' – although this is true, I suggest rewording, such as, 'k was found to be independent of size' since several other studies have found k to be size–dependent.

We corrected the text as suggested.

L85 – insert 'geographic' before 'differences persist: : :'

We corrected the text as suggested.

L106–108 – It would be worth saying why models still assume the same dust composition globally – e.g. due to computational cost of additional tracers? And/or lack of a globally consistent information dataset?

Following the reviewer suggestion we modified the text as below:

*"In spite of this sensitivity, present climate models adopt a globally–constant spectral complex refractive index (and SSA) for dust, and hence still implicitly assume the same dust mineralogical composition at the global scale. This is mainly due to the lack of a globally consistent dataset providing information of the geographical variability of the dust scattering and absorption properties (e.g., Sunset et al., 2018)."*

L207 – 'Ogreen' typo

We corrected the typo as suggested.

L257 – RI range for n and k – which values were used? Different values for different experiments? Where/how is this range applied?

As explained in Di Biagio et al. (2017) the optical to geometrical diameter conversion was performed as following: optical calculations were computed over the spectral range of the WELAS and at the GRIMM operating wavelength using Mie theory for spherical particles by fixing n at 1.47, 1.50, and 1.53, and by varying k in steps of 0.001 between 0.001 and 0.005. Dg was then set at the mean ±1

standard deviation of the values obtained for the different n and k. This calculation procedure is now better described in Table 1 where all the details of instrument artefact corrections are provided.

L268 – 'cut at 10 microns' – this contradicts p5 l178 which says 5 microns/8 microns (50/100% efficiency).

There is no contradiction since the 10 µm and the 8 µm represent respectively the 100% cutoff for the SW optical instruments and the OPCs sampling lines due to their geometry. As explained in the text, the particle losses along sampling lines for OPCs were corrected to retrieve the "real" size distribution of the dust aerosols in CESAM, $dN/dlogD_{CESAM}$. Successively the losses along sampling lines for SW instruments were accounted for to derive from $dN/dlogD_{CESAM}$ the size distribution sensed by SW optical instruments that is denoted in the text as $dN/dlogD_{SWoptics}$.

L278–279– It would be useful to mention the cut–off diameters again here as deff,coarse does not represent the full size range.

We clarified this point in the text as suggested by the reviewer.

L280 – it would be useful to say why modes were fitted to the size distribution, for non–experts.

We modified the sentence as below:

*"The dust size distribution, $(dN/dlogD)_{SWoptics}$, measured at each 10–min time step for each sample was fitted with a sum of five lognormal functions to smooth data inhomogeneities linked to the different instrument's operating principles and artefacts".*

L283 – how were modes fitted? (I think this is provided later in the paper but it should be mentioned here).

We added this sentence in the text to specify how the fitting was performed:

*"Fitting was performed using the Levenberg–Marquardt algorithm".*

L301 – what about other uncertainties to the size distribution, such as shape assumptions and/or Mie–regime singularities?

Concerning the uncertainties in the spherical assumption, we extended the discussion in Sect. 3.3 and we remind to it at the end of Sect. 2.2. Concerning the uncertainties in Mie regime singularities, basically they come from the fact that if optical calculations are performed at too low size resolution then it is not possible to resolve Mie singularities. We tried to test the impact of that on our results by performing Mie calculations at higher diameter resolution (dlogD=0.01 µm) than used in original calculations (dlogD=0.06 µm). We found that increasing the resolution of the calculation modifies less than 1% on average our results for the optical to geometrical diameter correction. So in our case the impact of Mie singularities is almost negligible. We discussed these two points in the main text.

*"Other sources of uncertainties are linked to the spherical assumption to perform the optical to geometrical diameter conversion (discussed in Sect. 3.3) as well as those due to Mie resonance oscillations of the calculated scattering intensities. Concerning Mie resonances, a sensitivity study was performed varying the size resolution of our calculations (high/low diameter resolution in the calculations to have a better/worse reproduction of Mie resonance oscillations) and show that Mie resonances impact the optical to geometrical correction by less than 1%."*

L342–3 – and also when comparing the RI data?

The refractive index dataset is shown not to be sensitive to changes in the size distribution, based on the results of our analysis. Therefore differences in temporal sampling for the different samples

should not affect the comparison of data for the refractive index. Nonetheless, and in order to keep generality, we rewrote the final sentence of the section as:

"*This difference in time sampling should be kept in mind when comparing data for these two samples to the rest of the dataset.*".

L402 – 'contrasting' ! 'contradictory'?

We corrected the text as suggested.

L447 – Muller et al (2011) report observations at Capo Verde, not transported across the Atlantic.

The reviewer is right, so we reformulated the sentence as below

"*Conversely, the values of $D_{eff,coarse}$ behind the SW instruments inlets are mostly in agreement with those reported for dust transported at Capo Verde and across the Atlantic ocean (~3 µm, Maring et al., 2003; Müller et al., 2011; Denjean et al., 2016b)*".

L460–464 – And also the fact that the size distribution above the 50% transmission efficiency (5 microns?) is not well represented, should be mentioned.

This point is now specified in the paper. The new text is:

"*The overall shape of the dust size distribution sensed by the SW instruments is comparable to that measured during atmospheric long–range transport, even if the fraction of particles above 3.9 µm diameter, which is at the 50% cutoff of the transmission efficiency for the SW optical instruments, is significantly under–represented compared to observations. (i.e., Betzer et al., 1988; Formenti et al., 2001; Maring et al., 2003; Ryder et al., 2013b; Jeong et al., 2014; Denjean et al., 2016b). »*

Figure 4 and discussion in lines 451–464 – the authors should consider that some of the observational data they show from other campaigns was also restricted by maximum size measured or by inlet transmission efficiencies (e.g. NAMMA, PRIDE). Information on some of these restrictions are provided in Ryder et al. (2018), table 1. As such, some of these datasets likely underestimate the coarse mode size distribution. Transported dust size distributions are also available for the AER–D campaign in the same paper which would add to the data already shown in Figure 5 and did not suffer from inlet restrictions.

We added data from AER–D (Ryder et al., 2018) and from SALTRACE (Weinzierl et al., 2017) in Figure 4. We also added a brief discussion on the inlet restriction for field data at the end of Sect. 4.1.1:

"*It should be keep in mind that often also field data are affected by inlet restrictions so that they cannot measure the whole coarse dust fraction (see Table 1 in Ryder et al., 2018). The lowest cutoff for field data shown in Fig. 4 are for the NAMMA and PRIDE datasets and correspond to upper size limits at 5 and 10 µm in diameter, respectively. Being these values above our cutoff of 3.9 µm, it means that the comparison with our size dataset is meaningful within the range of our measurements. To note that only the data from AER–D did not suffer from significant inlet restrictions thus leading to the observation of giant dust particles up to tens of microns in the Saharan Air Layer off the coasts of Western Africa.*"

L466 – it would be useful to add a line on the importance of iron oxides vs elemental iron for the benefit of non–specialists.

We included this sentence in the main text:

"*Elemental iron include the iron in the form of iron oxides and hydroxides, hematite and goethite (the so–called free iron, mostly controlling SW absorption) and the iron incorporated in the crystal structure*

*of silicates and alluminosilicates (illite, smectite), conversely not considerably contributing to SW absorption (Karickhoff and Bailey, 1973; Lafon et al., 2004).".*

L469 – 'Australia' – should this be Namib–2? (values are not consistent with those in the table).

The reviewer is right and we corrected the text accordingly.

L469 – 'Iron oxides account for 11 and 62% of the iron mass' – where do these values come from? They are not shown in table 3?

The fraction of the iron mass that is in the form of iron oxides was determined by XANES (X–ray absorption near–edge structure) measurements as described in more detail in Caponi et al. (2017). Briefly, the XANES spectra of the dust sample was deconvoluted using the spectra of five standards of Fe(III)–bearing minerals previously measured with the same technique. The linear deconvolution provided the proportionality factors representing the mass fraction of elemental iron to be assigned to the $i_{th}$ standard mineral. In particular, the values of the proportionality factors for hematite and goethite represent the mass fractions of elemental iron that can be attributed to these two species. The mass of the Fe oxides is the sum of the mass fractions of hematite and goethite, assumed to be the mass fraction of elemental iron that can be attributed to iron oxides. We specified in the text that this values was calculated following Caponi et al. (2017) so that the reader can refer to this paper for further details.

Section 4.1.2 – did iron content vary with particle size? Was that measured?

No, we did not measure the iron oxide content as a function of the particle size during experiments but only for the fraction corresponding to PM10.6 aerodynamic diameter as discussed in Caponi et al. (2017) and now specified in the main text.

L487 – angstrom exponents across which wavelengths?

The Ångstrom Exponents were calculated over the entire 370–950 nm spectral ranges and this is now specified in the main text.

L549 – Steigmann – typo either here or in references

We corrected the typo here and also somewhere else in the text and figures.

L527 – this is a very long paragraph and would benefit from being broken up a bit.

We modified the discussion in Sect. 4.3 and broke up the paragraph as suggested by the reviewer.

L557–560 – if this is the case regarding the Wagner dataset, wouldn't that make their dataset more reliable?

We extended the discussion regarding the differences with the Wagner et al. dataset in Sect. 4.3. First in Figure 8 we now provide both the max (Burkina Faso) and the min (Cairo) of the k spectra reported by Wagner et al. (2012). This permits to show that the main outlier is the Burkina Faso sample but that also the min k by Wagner et al. (Cairo sample) is at the upper edge of our data. We now fully discuss the differences between our laboratory experiments and the one by Wagner so to underline that apart from the shape assumption, there are other factors possibly leading to the observed differences (generation technique, size distribution, assumptions on n in the retrieval). Even if it is difficult to assess which is the dominant source of discrepancy, now the paper provides the reader the different elements to better evaluate the two different approaches.

The new text is reported in the following:

*"For the case of Wagner et al. (2012) the imaginary refractive index was retrieved from laboratory chamber experiments on suspended dust, as in our study. Nonetheless, their approach differs in various aspects from the one applied here and this can lead to the observed differences in the retrieved k. First, the aerosol generation technique is different between the two works and this possibly leads to particles with different physico–chemical features compared to our study. In Wagner et al. (2012) the dust aerosol was generated by a rotating brush disperser using only the 20–75 μm sieved fraction of the soils. This system acts to disaggregate the finest particles of the soil by passing it through a nozzle. Then the largest aerosol grains were removed by a cyclone system (50% cutoff at 1.2 μm aerodynamic diameter), so that only the submicron size fraction was measured. We show in Sect. 4.5 that k is independent of size for the range of investigated effective coarse diameters between 2 and 4 μm, but the range of sizes analysed in Wagner et al. (2012) is significantly lower than in our study and a size–effect cannot be excluded. In fact, the relationship between dust absorption and iron content may vary depending on the considered size fraction (see C17) due to the fact that iron bearing minerals are more concentrated in the clay fraction (<2.0 μm) of the dust (Kandler et al., 2009). Moreover, generating dust in a different way may lead to differences in the chemical and mineralogical size–dependent composition of the sample, therefore contributing to the observed differences. The impact of this is however difficult to evaluate. Another difference concerns the choice of the optical theory to retrieve k (T–matrix in Wagner et al. instead of Mie theory as used in our work). This can contribute to the observed differences, even if in a limited way (Mogili et al., 2007; Sorribas et al., 2015). Third, in their retrieval Wagner et al. fixed the real refractive index to a wavelength–independent value of 1.53 (as done in several other field and laboratory studies in Fig. 8) and this assumption can bias high/low the retrieved k if the actual n is higher/lower than the assumed 1.53 value. So, in summary, while multiple factors could contribute to the discrepancy it remains however difficult to assess which source of discrepancy is dominant."*

L544–560 – Comparing the new results to the older data is an interesting and important discussion point of the paper. It would be useful to expand this discussion to include a little more information on the methods of Wagner et al (2012) and Steigmann & Yang (2017) to allow the reader to better understand the different approaches. Are the authors able to justify that their method is more reliable than the other studies? This is done for the comparison to the Volz & Patterson work, but not the other mentioned publications. It would help readers if the authors are able to justify their results more strongly, as this will enable readers of the paper to use the new data over the older datasets with a high level of confidence, rather than just adding to the spread of existing data.

A more detailed discussion on the Wagner et al. (2012) dataset and reasons for discrepancies with our data is now provided in Sect. 4.3 (see also previous answer). Conversely, as previously discussed, the Stegmann and Yang (2017) dataset was not the correct one in the first version of our paper. The authors made a mistake in uploading the wrong data in the supplement to their paper and they provided us the correct data. These now fall within the range of our results. As we discuss more in detail only the literature data falling outside our range of results we consider it no more necessary in the revised text to discuss further the Stegmann and Yang (2017) dataset.

L569–580 – can the authors comment on the fit for hematite vs goethite? Based on these results, is it therefore necessary to measure *both* hematite and goethite (rather than only hematite, as is sometimes done), in order to retrieve appropriate absorption estimates?

Yes, since both hematite and goethite contribute to the iron oxide content in dust, as shown in Table 3 in our paper for example, and also both contribute to the absorption in the visible. A comment on this aspect was added in the text as below:

*"Measuring only the hematite mass fraction to estimate the dust absorption, as it is sometimes done, is therefore not sufficient."*

L589–593 – So is this several (or only one) data point per dust sample?

It is several data points per dust sample, since the 10 minutes average data are reported in the plot for each experiment.

L596–598 – is the same true if you plot deff_coarse vs iron oxide content? Wouldn't this be a more direct comparison?

The iron oxide content is integrated over the whole duration of each experiment while the deff,coarse is estimated at 10–min intervals, so the deff,coarse versus iron oxide content relationship would not add any information in this case due to the differences in their temporal resolution.

L599–601 'RI : : : is independent of size' – this is not always the case – e.g. Kandler et al. (2009) show that composition, and RI, change with size, and this is also reflected in the figures of Otto et al. (2009).

Based on the comments for both reviewers on this part of the discussion, we reformulated the paragraph as below:

*"These results confirm previous observations (Sokolik and Toon, 1999; McConnell et al., 2008, 2010; Ryder et al., 2013a; 2013b) that the refractive index is independent of size. This suggests that size–dependent mineralogical composition is not sufficient to affect k (in the limit of our measurement and retrieval procedure precision). It is worth mentioning that only few past studies evidenced a dependence of k on the size distribution of the dust aerosols (i.e., Kandler et al., 2009, 2011; Otto et al., 2009) maybe because the refractive index was retrieved in these studies from mixing rules based on the estimated size–dependent mineralogical composition."*

L604–605 – this was also found by Ryder et al. (2018) – i.e. SSA was dominantly dependent on composition.

We added the Ryder et al. (2018) reference to the main text:

*"Ryder et al. (2013a) also showed that the dependence of SSA on size is linear, but important only when the coarse fraction is high (if particles larger than about 3 µm in diameter are present), otherwise the SSA depends mainly on composition, also in agreement with more recent field observations by Ryder et al. (2018)."*

Section 4.5 – do the authors have any thoughts on how much, if at all, their RI data may apply (or differ) for dust close to the source, when the coarse mode is more prevalent? Would it be worth mentioning this again in the conclusions as a potential area for future research?

We already mentioned this aspect in the conclusions, nonetheless we modified the text in Sect. 4.5 in order to clarify this aspect.

*"Further investigation should be therefore addressed to evaluate the dependence of the spectral k and SSA versus iron content as a function of the size distribution of the particles, in particular extending to a wider range of $D_{eff,coarse}$ compared to the one investigated in the present study. This will allow to determine if the k and SSA versus iron relationships change or not in different phases of the aerosol lifetime, so if it is valid close to source areas (when the coarsest fraction is dominant, i.e. $D_{eff,coarse}$ up to 15 µm, Ryder et al. (2013b)), and in long–range transport conditions (when most of the coarse particle fraction above few µm has settled out, i.e. $D_{eff,coarse}$ of 2–3 µm or lower, Denjean et al. (2016b))."*

Summary & concluding remarks – Do the authors have any suggestions for how to extrapolate RI at wavelengths lower than 370nm or larger than 950nm? These wavelengths are often required for spectral data within GCMs and/or NWP. Or if not, perhaps this should also be mentioned as an area requiring work in the future.

Based on the results of previous studies (i.e., Wagner et al., 2012) we could suggest as a first approximation a linear interpolation of the CRI out of our investigated wavelength range. Nonetheless, limited information still exists for the UV part of the spectrum and above 1 µm wavelength, so further research is required. Following the reviewer suggestion we added this sentence in the conclusions:

*"Our study focus on the dust spectral optical properties between 370 and 950 nm. Further work is required to extend the range of spectral refractive index and SSA data at wavelengths lower than 370 nm or higher than 950 nm given that these data are often required within Global Circulation Models and Numerical Weather Prediction models."*

L625–627 – Can the authors comment on the reason for their k data falling at the lower edge of the literature data, while SSA data seems very much in the middle of the literature range? This might be considered somewhat contradictory.

Our k falls at the lower edge of the ensemble of literature data but looking at Fig. 8 it is possible to see that our k values fully envelope the ensemble of remote sensing and field campaign data on airborne dust, which are the same sources of SSA data. So, there isn't contradiction between the k and SSA data in our opinion. The k data that are above our estimates are the one of Volz (1972), Patterson et al. (1977), Hess et al. (1998) and Wagner et al. (2012) discussed in the main text. We slightly modified the text in Sect. 4.3 to clarify this point.

L646–648 – this sentence contradicts the discussion around figure 9 where the authors say that the coarse mode (deff) impacts the SSA – clearly the size distribution impacts the SSA too.

The reviewer is right and we reword points 3 and 4 in the conclusions, and added a summary point 5 in order to take this into account. We also pointed out in the conclusions on the necessity to have a good representation of the dust coarse mode size distribution in models given its crucial impact on the particle's SSA.

The new text reads as follows:

3. *"The sample–to–sample variability observed in this study is mostly related to the iron oxide and elemental iron content in dust. At each investigated wavelength the magnitude of k and SSA is linearly correlated to the mass concentration of total iron oxides, hematite, goethite, and total elemental iron. Small variations of these compounds translate into large variations of k and SSA.*
4. *We also investigated the dependence of k and SSA on the size distribution of dust. While k is independent of size (suggesting that a constant value can be used along transport), below 600 nm the SSA linearly decreases for increasing $D_{eff,coarse}$ for strongly absorbing samples with more than 3% iron oxide content. The investigated range of $D_{eff,coarse}$ is within about 2 and 4 µm, and thus comparable to values obtained along a transport path over the Atlantic Ocean for dust during about 2 to 6 days following emission (Denjean et al., 2016a).*
5. *The observations of points 3 and 4 suggest that while it is sufficient to know the content of iron oxide (or elemental iron) in dust to predict its spectral k, which means that only one tracer is needed in models to parametrize its regional and global variability, for the spectral SSA both composition and size distribution are required."*

L657 – 'we propose: : : a set of regionally–averaged n, k and SSA values: : :' I wonder how useful this would really be – e.g. RI values are severely different for several samples in the same region – e.g. the Sahel, South Africa, East Asia (fig 6) so that even a regional representation may underestimate the sub–regional variability. Additionally there is the computational cost of additional tracers that may be required. The authors should mention some of these issues.

We agree with the reviewer that the regionally–averaged values provide only a partial representation of the dust k and SSA variability. Nonetheless, giving average and standard deviation values for each of the investigated regions is a starting step to go toward a better geographical characterization of the dust optical properties. Indeed we mention this aspect in the last paragraph of the paper that we

report below, also discussing the importance of the regional and sub–regional variability emerging from our dataset and the need to investigate it. Together with regional averages we also provide in the paper the full set of original data of refractive indices and SSA for the 19 analyzed samples (Table 4 and 5). Users are welcomed to use the original data instead of regional averages.

Concerning the computational cost of additional tracers, we mostly developed a discussion in the conclusion section concerning the need of having a reliable mineralogy of dust in models, especially iron oxides, as a good tracer of dust absorption properties.

*"Finally, this study had the objective to investigate the variability of the dust SW optical properties at the global scale linking to the global variability of the dust composition. It is noteworthy that observations over Southern Africa and the Sahel from the present study indicate that the k and SSA variability over these regions is comparable to the one obtained for the global scale. For other regions, such as North America and Australia, only one sample was analyzed, with no information on the regional–scale variability of k and SSA. Additionally, for some of the analyzed areas, such as the Bodélé depression, even local scale variability (on the order of few km) may be of relevance, given the documented local scale changes of the particles' mineralogy and iron content (Bristow et al., 2010). More efforts should be therefore devoted to better characterize the variability of dust spectral optical properties at the regional and sub–regional scale with the aim of better assessing the dust impact on the climate of different areas of the world."*

678 – 'in link' ! linking?

We followed the suggestion of the reviewer and modified accordingly in the text.

Tables & Figures

Table 3 caption +/–10%, 15%, 10% – what exactly are these uncertainties? They seem very large for % values in the table of _1–10%?

The values of 10% and 15% are relative uncertainties and not absolute uncertainties. We specified this point in Table 1 and 3.

Figure 2 caption – it's not quite clear which data is which – dots are Journet data, so are the box+whiskers Ginoux/DB17? Also, if the current work uses the same samples, shouldn't the iron oxide data be the same?

In Figure 2 the box and whisker and the dots are both extracted from Journet et al. (2014) providing data for clay–sized and silt–sized soil fractions. The box and whisker represents large geographical areas corresponding to the global dust sources indicated in Ginoux et al. (2012) and used in DB17, whereas dots are extracted corresponding to the coordinates where the nineteen soils used for CESAM experiments were collected. Both data in the plot refer to the soil. Differently, the iron oxides data for the CESAM samples analyzed in this study refer to the aerosols phase. Therefore the two datasets (dots in Figure 2 and data for iron oxides in Table 3) are not the same.

We modified the caption of Fig. 2 following the reviewer's remark in order to clarify it. The new caption reads as below:

*"Box and whisker plot showing the full variability of hematite and goethite mass fractions in the soils for the clay–sized (<2 μm diameter) and silt–sized (<60 μm diameter) fractions as retrieved from the global soil mineralogical database by Journet et al. (2014). The box and whisker plot include data for the nine desert source areas depicted in Ginoux et al. (2012) and DB17 (Northern Africa, Sahel, Eastern Africa and the Middle East, Central Asia, Eastern Asia, North America, South America, Southern Africa, and Australia). Dots indicate hematite and goethite content in clay–sized and silt–sized soils (always from Journet et al.) extracted in correspondence of the geographical coordinates where the nineteen soils used in the CESAM experiments were collected. The Journet et al. database assumes that the iron oxides in the silt fraction consist only of goethite."*

Figure 3 – x–axis label – 'time (hours)' – given this starts at @1330, is this time of day? Time from start of experiment would be more appropriate.

We modified the x–axis in order to represent the time in hours from the dust injection in CESAM (the t=0 is the moment when the dust starts to be injected in the CESAM chamber).

Figure 6 – these panels are all far too small to interpret. These should be made significantly larger. I also suggest changing the axis ranges so that the data takes up more than a minimal fraction of the plot.

We agree with the reviewer that the plot was difficult to read so we decided to split it in two, one figure for the real and imaginary parts of the refractive index (Figure 6) and one for the SSA (new Figure 7). Concerning the y axes we tried to enlarge the scale as much as possible but keeping the same scale for all the different plots in order to evidence differences between the different datasets. We hope the new version of the plots is more readable.

Figure 7 caption – L1256 'measured' – I suggest using 'represented' instead as for several of these studies the properties shown (RI, SSA) are not directly measured, but calculated from other more directly measured properties.

We corrected as suggested by the reviewer.

Figure 7 – These panels & data displayed are far too small to interpret. I suggest removing the left hand column (data from this study) since that is replicated on the right hand column and can be seen here. The literature data is virtually impossible to interpret. I suggest making all these plots much larger and also expanding the y–axis range for the k and n plots.

We modified Fig. 7 as suggested by the reviewer in order to improve its readability. To note that the dataset from Stegmann and Yang (2017) was modified in this revised figure due to an error in their published data. Corrected data were provided to us by the authors and these new values are now showed in the plot.

Figure 9 – This figure could also do with the plots being larger, and having more zoomed–in displays of the data.

We modified Fig. 9 in order to enlarge as much as possible the scales but maintaining all the relevant information (notably error bars) included in the plots. We increased also the fonts and rearranged the legends in order to make them more readable.

**Carlos Pérez García–Pando (Referee #2)**

The paper by Di Biagio et al. presents a new dataset of complex refractive indices and single scattering albedo (SSA) at 7 wavelengths in the shortwave range for dust aerosols generated from 19 soil samples collected from the 8 more prominent dust emission regions on Earth. The dataset was derived after resuspending dust into a smog chamber and measuring its spectral scattering and absorption coefficients and size distribution. The complex refractive index was estimated by Mie calculations combining the optical and size data, and the SSA was derived directly from the scattering and absorption coefficients. In addition, bulk composition was measured for each sample to better understand the influence of mineralogy upon absorption. A main result is that the retrieved imaginary part (k) is in the lower range of values currently found in the literature. The study also provides wavelength–dependent (strong) linear relationships between k and SSA, and the content of iron oxides that may help simplifying the implementation of the effect of dust mineralogy upon absorption in climate models. This well–written paper provides a new and unprecedented dataset along with new insights on the dependencies of the absorption on the content of iron oxides. Therefore, I strongly recommend its publication after addressing some minor general and specific comments that can be found below.

General Comment

The authors make several assumptions in the optical calculations in order to retrieve k from the measured data. These assumptions are clearly summarized in section 3.3. First, the size distribution from the OPC and the scattering coefficient depend upon the refractive index; the approach was to set fixed values. Second, the OPC optical–to–geometrical diameter conversion and the nephelometer truncation correction (in addition to the retrieval algorithm in section 3.2) assume homogeneous spherical particles and therefore use Mie theory. In the first case the authors calculate the sensitivity of the results to the assumption and demonstrate the robustness of their approach. However, the potentially large impact of the second assumption (shape) is not quantified. It is argued that shape was not measured and that improper assumptions on the particle shape or morphology may induce even larger errors. I find this last argument a little weak: 1) There are estimates of shape in the literature and recent studies (e.g. Kok et al., 2017) highlighting the potential importance of shape upon extinction (using triaxial spheroids); 2) the influence of shape upon the conversion from optical to geometrical diameter seems also quite important (increasing the amount of fine particles compared to coarse ones; Huang and Kok, 2018); 3) The nephelometer truncation error should also be quite sensitive to shape; 4) The authors compare their results with other estimates in the literature; in page 15 they argue that the strong differences with Wagner et al. (2012) may be due to the choice of the optical theory (T–Matrix vs Mie), which somehow admits that assuming spherical particles may be a strong source uncertainty.

While I understand that accounting for shape is a complex issue beyond the scope of this paper, I nevertheless ask the authors to more clearly acknowledge this (potentially strong) uncertainty both in the abstract, discussion (section 3.3) and conclusions. I would also invite the authors to explore and report on this uncertainty in future studies.

We thank the reviewer for these comments. We added some text in Sect. 3.3 to better discuss the question of the shape assumption, providing also additional discussion in the comparison against Wagner et al. (2012). First, it should be noted that there are studies showing that while a non–spherical shape modifies the phase function, the differences between spherical and non–spherical approaches become less important when calculating angular–integrated quantities as it is the case of integrated scattering, or SSA, and the asymmetry factor (Mishchenko et al., 1995; Otto et al., 2009). For OPC optical to geometrical correction, nephelometer truncation correction and also for n and k retrieval basically optical calculations are used to calculate angular–integrated quantities, so the impact of shape can be assumed to be moderate. For instance, Sorribas et al. (2015) showed that using a spheroidal model has a negligible effect in the truncation correction. Nonetheless, it is difficult

to quantify the overall uncertainty introduced in our results by shape assumptions and this for two main reasons. First, because measurements of the dust shape distribution from field data exist, as also indicated by the reviewer, but these show that dust can present very complex, irregular, and variable shapes, so in the absence of measurements for the actual analyzed samples it is very hazardous to associate a shape to our dust aerosols. Second, even if a shape is assumed then additional questions arise concerning the optical theory to use to model it. Indeed, given the complexity of dust shape there is still debate about which is the best theory to represent its optical properties in particular as a function of the different envisaged applications (see Nousiainen and Kandler, 2015). Nonetheless we agree with the reviewer and we acknowledge the need for further investigation of the impact of shape on our results. We address this future field of research in the conclusions by adding the following text:

"*We do not provide any quantification of the uncertainty associated with the assumption of spherical particles in our study, even if we acknowledge the potential role of non–sphericity in affecting our data treatment and results. Additional work is foreseen to better investigate the shape of our generated dust and the impact of non–sphericity on retrieved spectral refractive indices and SSA.*"

We now also mention more clearly in the abstract the fact that the spherical assumption for dust may introduce a potentially source of uncertainty:

"*Dust is assumed to be spherical in the whole data treatment, which introduces a potential source of uncertainty.*"

Specific comments

Title: The current form of the title may induce to confusion (it is not clear whether size relates to iron size or dust size). I suggest two possibilities: "Complex refractive indices and single scattering albedo of global dust aerosols in the shortwave spectrum and relationship to size and iron content" or "Complex refractive indices and single scattering albedo of global dust aerosols in the shortwave spectrum and relationship to iron content and dust size"

We understand the remark of the reviewer and the title can be changed following the first reviewer's suggestion, that is:

"*Complex refractive indices and single scattering albedo of global dust aerosols in the shortwave spectrum and relationship to size and iron content*"

Abstract: Please emphasize that the linear relationship is stronger with iron oxides than with total iron. Also, indicate that your estimates are in the lower range of the literature values.

The abstract was modified to include this information.

L197 to L217.: Please elaborate a bit on the assumption of spheres for the truncation correction and please connect it to an improved discussion in section 3.3 as suggested above.

We extended the discussion in Sect. 3.3 concerning the spherical approximation. Also we cite here the results of the study by Sorribas et al. (2015) that tested the impact of spherical versus spheroidal models in nephelometer truncation corrections in modelling of the scattering coefficient. They showed that using a spheroidal model has a negligible effect in the truncation correction. The new text is reported in the following:

"*The retrieval procedure for n and k, as well as the calculations for OPCs optical–to–geometrical diameter and the nephelometer truncation correction, simplifies the non–spherical heterogeneous dust aerosols (e.g., Chou et al. 2008; Okada et al., 2011; Nousiainen and Kandler, 2015) into homogeneous spherical particles that can be represented by Mie theory. In the present study, we decided not to use a more advanced shape–representing theory for three main reasons. First, the spherical model has been shown to produce only moderate errors when computing angular–integrated quantities (Mishchenko et al., 1995; Otto et al., 2009; Sorribas et al., 2015) such as those*

*we calculate in this study to retrieve the OPC and truncation corrections and for n and k retrieval. For instance, Sorribas et al. (2015) showed that using a spheroidal model has a negligible effect on the truncation correction. These authors estimated that using a spheroidal model permits to improve by 4 to 13% the agreement between modelled and measured spectral scattering coefficient at 450–700 nm but only for supermicron particles. Conversely, for submicron dust the spherical approximation is better suited than the spheroidal model to reproduce the scattering coefficients by the nephelometer. The study by Mogili et al. (2007) also found an excellent agreement between measured shortwave extinction spectra and those calculated from Mie theory simulations for dust minerals, supporting the use of Mie theory for dust optical modelling. On the other side, other studies point to the need of a non–spherical assumption to improve the modelling of dust optical properties (e.g., Otto et al., 2009). Second, we used Mie theory for the sake of comparison with the large majority of previous field and laboratory data published so far, which had used calculations with the spherical approximation Third, the shape distribution and morphology of the dust samples was not measured during experiments. Improper assumptions on the particle shape and morphology may induce even larger errors than using Mie theory, in particular for super–micron aerosols (Kalashnikova and Sokolik, 2004; Nousiainen and Kandler, 2015). It should be pointed out, however, that dust is usually assumed to be spherical in global climate models (e.g., Myhre and Stordal, 2001; Balkanski et al., 2007; Jin et al., 2016), and different studies still show contradictory results on the true impact of dust non–sphericity on radiative fluxes and heating rates from global model simulations (Mishchenko et al., 1995; Yi et al., 2011; Räisänen et al., 2012; Colarco et al., 2014). On the other hand, shape effects can be important for the retrieval of aerosol properties from remote sensing techniques using spectral, angular, and polarized reflectance measurements (e.g., Feng et al., 2009). In synthesis, accounting for shape effects is still controversial for dust modelling and also a complex issue beyond the scope of this paper. Thus, while we acknowledge the potential uncertainties induced by spherical assumptions in our study, we do not quantify here the overall impact of this assumption on our results."*

L241: "varied between: : :. nm and: : :."

We corrected as suggested by the reviewer.

L268: 10 microns?

We corrected as suggested by the reviewer.

L301: what about shape?

We added some text here and we developed Sect. 3.3 in order to better discuss the spherical assumption in our analysis.

L542: what do you mean by "our" population average of 1.8 % : : :.?

Following the reviewer's remark, we changed the paragraph as below:

*"Likewise, our k average is also very close to the dataset by Balkanski et al. (2007), estimated from mineralogical composition assuming 1.5% (by volume) of hematite in dust, a value shown to allow a reconciliation of climate modelling and satellite observations of the dust direct SW radiative effect. By comparison, the average dust hematite content for the ensemble of our analysed samples is 1.8% (in mass), close to the 1.5% value proposed by Balkanski et al. (2007)."*

L557: Please develop more. This is quite important in my opinion as I highlighted above. Can you further discuss the differences with Wagner?

We extended the discussion regarding the differences with the Wagner et al. dataset in Sect. 4.3. First in Figure 7 we now provide both the max (Burkina Faso) and the min (Cairo) of the k spectra reported by Wagner et al. (2012). This permits to show that the main outlier is the Burkina Faso

sample but that also the min k by Wagner et al. (Cairo sample) is at the upper edge of our data. We now fully discuss the differences between our laboratory experiments and the one by Wagner so to underline that apart from the shape assumption, there are other factors possibly leading to the observed differences (generation technique, size distribution, assumptions on n in the retrieval). Even if it is difficult to assess which is the dominant source of discrepancy, now the paper provides the reader the different elements to evaluate the two different approaches.

The new text reads as follows:

*"For the case of Wagner et al. (2012) the imaginary refractive index was retrieved from laboratory chamber experiments on suspended dust, as in our study. Nonetheless, their approach differs in various aspects from the one applied here and this can lead to the observed differences in the retrieved k. First, the aerosol generation technique is different between the two works and this possibly leads to particles with different physico–chemical features compared to our study. In Wagner et al. (2012) the dust aerosol was generated by a rotating brush disperser using only the 20–75 μm sieved fraction of the soils. This system acts to disaggregate the finest particles of the soil by passing it through a nozzle. Then the largest aerosol grains were removed by a cyclone system (50% cutoff at 1.2 μm aerodynamic diameter), so that only the submicron size fraction was measured. We show in Sect. 4.5 that k is independent of size for the range of investigated effective coarse diameters between 2 and 4 μm, but the range of sizes analysed in Wagner et al. (2012) is significantly lower than in our study and a size–effect cannot be excluded. In fact, the relationship between dust absorption and iron content may vary depending on the considered size fraction (see C17) due to the fact that iron bearing minerals are more concentrated in the clay fraction (<2.0 μm) of the dust (Kandler et al., 2009). Moreover, generating dust in a different way may lead to differences in the chemical and mineralogical size–dependent composition of the sample, therefore contributing to the observed differences. The impact of this is however difficult to evaluate. Another difference concerns the choice of the optical theory to retrieve k (T–matrix in Wagner et al. instead of Mie theory as used in our work). This can contribute to the observed differences, even if in a limited way (Mogili et al., 2007; Sorribas et al., 2015). Third, in their retrieval Wagner et al. fixed the real refractive index to a wavelength–independent value of 1.53 (as done in several other field and laboratory studies in Fig. 8) and this assumption can bias high/low the retrieved k if the actual n is higher/lower than the assumed 1.53 value. So, in summary, while multiple factors could contribute to the discrepancy it remains however difficult to assess which source of discrepancy is dominant."*

L600: k is an intrinsic property of matter by definition. Perhaps rephrase this sentence. k is independent of size if mineral composition does not change with size.

Following the reviewer suggestion, we eliminated "as an intrinsic property of matter" from the sentence. Also, based on the comments for both reviewers we reformulated the paragraph as below:

*"These results confirm previous observations (Sokolik and Toon, 1999; McConnell et al., 2008, 2010; Ryder et al., 2013a; 2013b) that the refractive index is independent of size. This suggests that size–dependent mineralogical composition is not sufficient to affect k (in the limit of our measurement and retrieval procedure precision). It is worth mentioning that only few past studies evidenced a dependence of k on the size distribution of the dust aerosols (i.e., Kandler et al., 2009, 2011; Otto et al., 2009) maybe because the refractive index was retrieved in these studies from mixing rules based on the estimated size–dependent mineralogical composition."*

L642: I would rephrase this sentence. The variability is controlled by the iron oxide content. Using total iron partly reflects this because oxides are a large fraction of it but the linear relationship is obviously not as strong.

We prefer leave the sentence in the present form as there is not an obvious consensus on this aspect. Actually, several laboratory studies found strong correlation of shortwave absorption

properties also with elemental iron, sometimes also stronger correlations with elemental iron than with iron oxides (Caponi et al., 2017; Moosmuller et al., 2012; Engelbrecth et al. 2016).

L650: This sentence is a little confusing. k is independent of size by definition. However, an "effective" k of a size distribution may change if mineralogy changes with size and size evolves with transport. In fact in L667 you further propose to investigate this potential dependence.

The reviewer is right and we tried to clarify this aspect in the conclusions.

L657: the variability within regions can be very large though, please highlight that.

A discussion on the large regional variability evidenced in the present study is already provided in the last paragraph of the conclusions.

L675: you may mention and cite EMIT (Green et al., 2018) as a potential near–future source of high resolution surface mineralogy data for arid and semi–arid regions

We now mention the EMIT project as a potential source of new data on the surface mineralogy in the conclusions of our paper.

Figures: Figure 1 is not readable, please increase the font size in the flowchart

Figure 1 was modified to be more readable.

References:

Green, R. O.; Mahowald, N. M.; Clark, R. N.; Ehlmann, B. L.; Ginoux, P. A.; Kalashnikova, O. V.; Miller, R. L.; Okin, G.; Painter, T. H.; Pérez García–Pando, C.; Realmuto, V. J.; Swayze, G. A.; Thompson, D. R.; Middleton, E.; Guanter, L.; Ben Dor, E.; Phillips, B. R. (2018) NASA's Earth Surface Mineral Dust Source Investigation. American Geophysical Union, Fall Meeting 2018, abstract #A24D–01

Huang, Y.; Kok, J. F. (2018) Harmonization of size distributions of dust at emission. American Geophysical Union, Fall Meeting 2018, abstract #A21I–2821

Kok, J.F., D.A. Ridley, Q. Zhou, R.L. Miller, C. Zhao, C.L. Heald, D.S. Ward, S. Albani, and K. Haustein, 2017: Smaller desert dust cooling effect estimated from analysis of dust size and abundance. Nature Geosci., 10, no. 4, 274–278, doi:10.1038/ngeo2912.

---

## Author Comment (AC2) · 11 Oct 2019

[revised manuscript text omitted]

---

## Author Comment (AC3) · 11 Oct 2019

The comment was uploaded in the form of a supplement:
https://www.atmos-chem-phys-discuss.net/acp-2019-145/acp-2019-145-AC3-supplement.pdf